EMBO
Molecular Medicine

# Mithramycin induces promoter reprogramming and differentiation of rhabdoid tumor

Maggie H Chasse[1], Benjamin K Johnson[1], Elissa A Boguslawski[1], Katie M Sorensen[1], Jessica E Rosien[2], Min H Kang[3], C Patrick Reynolds[3], Lyong Heo[1], Zachary B Madaj[1], Ian Beddows[1], Gabrielle E Foxa[1], Susan M Kitchen-Goosen[1], Bart O Williams[1], Timothy J Triche Jr[1] & Patrick J Grohar[1,4,5,*]

## Abstract

Rhabdoid tumor (RT) is a pediatric cancer characterized by the inactivation of SMARCB1, a subunit of the SWI/SNF chromatin remodeling complex. Although this deletion is the known oncogenic driver, there are limited effective therapeutic options for these patients. Here we use unbiased screening of cell line panels to identify a heightened sensitivity of rhabdoid tumor to mithramycin and the second-generation analogue EC8042. The sensitivity of MMA and EC8042 was superior to traditional DNA damaging agents and linked to the causative mutation of the tumor, SMARCB1 deletion. Mithramycin blocks SMARCB1-deficient SWI/SNF activity and displaces the complex from chromatin to cause an increase in H3K27me3. This triggers chromatin remodeling and enrichment of H3K27ac at chromHMM-defined promoters to restore cellular differentiation. These effects occurred at concentrations not associated with DNA damage and were not due to global chromatin remodeling or widespread gene expression changes. Importantly, a single 3-day infusion of EC8042 caused dramatic regressions of RT xenografts, recapitulated the increase in H3K27me3, and cellular differentiation described *in vitro* to completely cure three out of eight mice.

**Keywords** EC8042; epigenetics; mithramycin; pediatric cancer; SWI/SNF
**Subject Categories** Cancer; Chromatin, Transcription & Genomics

## Introduction

The SWI/SNF chromatin remodeling complex is mutated in ~ 20% of human cancers. Rhabdoid tumor (RT) is emblematic of this dysregulated complex and driven by biallelic inactivation of specific subunits, SMARCB1, or less commonly SMARCA4 (Versteege *et al*, 1998). Rhabdoid tumor can arise in the CNS (atypical/teratoid rhabdoid tumor), kidney (rhabdoid tumor of the kidney), or soft-tissues (malignant rhabdoid tumor). Regardless of tumor location, patients are treated with multimodal therapy consisting of high dose chemotherapy, surgery, and/or radiation (Tekautz *et al*, 2005). While surgery, radiation, and chemotherapy are options for RT, the four-year survival is still 10–40%, with prognostic outlook worsening with a younger age at diagnosis (Guidi *et al*, 2020; Hoffman *et al*, 2020; Reddy *et al*, 2020). Further, the multimodal treatment regimen is associated with significant acute and chronic toxicities (Le Deley *et al*, 2005; Saad & Wang, 2015). Therefore, a great need exists for more effective and less toxic targeted therapy.

The deletion or inactivation of SMARCB1 lowers the affinity of SWI/SNF for chromatin, leading to a redistribution of the complex in the genome and aberrant activity (Alver *et al*, 2017; Nakayama *et al*, 2017; Michel *et al*, 2018; Erkek *et al*, 2019; Wang *et al*, 2019). Here, SWI/SNF occupancy alters gene expression to drive proliferation and block differentiation by altering the expression of multiple pro-oncogenic pathways such as Notch, Hedgehog, TGFβ, and Wnt β-catenin (Wang *et al*, 2009; Erkek *et al*, 2019). Importantly, recent studies have defined a non-canonical SWI/SNF (ncSWI/SNF) activity as a synthetic lethal target in rhabdoid tumor (Michel *et al*, 2018; Wang *et al*, 2019). Furthermore, these studies identified occupancy of ncSWI/SNF at promoters due to an interaction with CTCF (Michel *et al*, 2018). This interaction of ncSWI/SNF with CTCF perhaps directs dysregulated promoter enhancer interactions that have previously been described in rhabdoid tumor (Alver *et al*, 2017; Ren *et al*, 2017; Wang *et al*, 2017). In addition, the weakened affinity of SWI/SNF for chromatin disrupts antagonism with polycomb repressive complexes (PRC) (Kia *et al*, 2008; Wilson *et al*, 2010; Kadoch *et al*, 2017). This antagonism provides the basis for EZH2 targeting and the development of EZH2 small molecule inhibitors (Knutson *et al*, 2013). Here, we propose a complementary approach of directly targeting the SMARCB1-deficient SWI/SNF with mithramycin or its second-generation analogue EC8042. The goal is to reverse the

1 Van Andel Research Institute, Grand Rapids, MI, USA
2 Dartmouth College, Hanover, NH, USA
3 Texas Tech University Health Sciences Center, Lubbock, TX, USA
4 The Children's Hospital of Philadelphia, Philadelphia, PA, USA
5 University of Pennsylvania, Perelman School of Medicine, Philadelphia, PA, USA
 *Corresponding author. Tel: +1 267 425 0494; E-mail: groharp@email.chop.edu

expression of pro-oncogenic pathways and restore the differentiation program.

In this study, we show that the small molecule mithramycin and its second-generation analogue EC8042 inhibit SWI/SNF activity. We previously identified mithramycin as an inhibitor of the driver oncogene of Ewing sarcoma, EWS-FLI1 (Grohar *et al*, 2011). As part of these studies, we noticed a hypersensitivity of rhabdoid tumor to mithramycin and EC8042 (Osgood *et al*, 2016). In this report, we confirmed these results with another cell line screen, analysis of an independently generated dataset, and validate the results (Teicher *et al*, 2015). We link the observed hypersensitivity of RT to SMARCB1-deficient SWI/SNF activity. We show mithramycin displaces SWI/SNF from chromatin to cause focal chromatin remodeling, loss of CTCF accessibility to restore PRC2 activity and H3K27me3. These effects trigger epigenetic reprogramming that favors promoters to block numerous oncogenic pathways associated with rhabdoid tumor and induce differentiation of the tumor cells into benign mesenchymal tissue both *in vitro* and *in vivo*. Importantly, the cellular sensitivity is not due to non-specific DNA damage, general transcription inhibition, or global changes in chromatin structure. The net effect is durable regression in rhabdoid tumor xenografts and completely cures of a subset of mice treated with a single, 3-day infusion of the mithramycin analogue EC8042. Because of the favorable toxicity profile of EC8042 (Osgood *et al*, 2016), these results establish EC8042 as a new clinical candidate for rhabdoid tumor patients.

# Results

## Rhabdoid tumor is sensitive to mithramycin, but not chemotherapy

We have previously identified mithramycin and its second-generation analogues as inhibitors of the oncogenic driver of Ewing sarcoma, the EWS-FLI1 transcription factor. As part of these studies, we screened a panel of pediatric cell lines and observed a striking sensitivity of rhabdoid tumor cell lines that paralleled Ewing sarcoma cells (see fig S5 in (Osgood *et al*, 2016)). We confirmed these results with a screen of an additional panel of pediatric cell lines and found two RT cell lines that were among the top 4 most sensitive cell lines to mithramycin out of 23 (Fig 1A). In order to show that these results were not an artifact of our screening, we analyzed a previously published cell line screen of sarcoma cell lines that was completed independent of our laboratory (Teicher *et al*, 2015). Again, we found RT was among the most sensitive cell lines, with Ewing sarcoma cell lines being the most sensitive to mithramycin (Fig 1B). To control for screening artifact, we selected a subset of cell lines representative of numerous histologies and performed rigorous dose–response curves. Again, we found RT cells were among the most sensitive to drug treatment with a relative sensitivity consistent with the screening data (Appendix Table S1). Importantly, the driver oncogene of Ewing sarcoma, the EWS-FLI1 transcription factor, associates with the SWI/SNF complex to direct the oncogenic program (Boulay *et al*, 2017). Therefore, the data suggested a common molecular feature associated with cellular sensitivity to mithramycin between Ewing sarcoma and rhabdoid tumor. Further, not every cell line was sensitive to the drug

suggesting a context dependence to activity. Indeed, SKOV3 ovarian carcinoma cells were more than 100× less sensitive to the drug with a micro-molar IC50 relative to either rhabdoid tumor or Ewing sarcoma cell lines that exhibited a low nanomolar IC50 (Appendix Table S1).

In order to link this sensitivity to common molecular features of Ewing sarcoma and rhabdoid tumor and exclude non-specific mechanisms of the drug, we compared the cellular sensitivity of these tumors to standard non-specific chemotherapeutic agents. Three RT cell lines (BT12, CHLA266, and G401) were 70-fold, fourfold, and 100-fold more resistant to three broadly active chemotherapy agents, etoposide, doxorubicin, and SN38 (the active metabolite of irinotecan) than Ewing sarcoma cells (Fig 1C). In contrast, we confirmed that all three cell lines were equally sensitive to mithramycin as the Ewing sarcoma cells, with a low nanomolar IC50 (Appendix Table S1). In addition, the effect was generalizable as four additional rhabdoid tumor cell lines showed comparable or increased sensitivity to mithramycin (Appendix Table S1). Importantly, these data reflect the clinical experience as Ewing sarcoma is known to be sensitive to these chemotherapeutic agents in the clinic (5-year event free survival 73%), while the overall response rate following high dose chemotherapy (methotrexate, vincristine, etoposide, cyclophosphamide, and cisplatin) in rhabdoid tumor is 43% (Womer *et al*, 2012; Reddy *et al*, 2020). Further, the data suggest that the mechanism of action of mithramycin leading to the heightened sensitivity of RT is likely unrelated to non-specific DNA damage, as the RT cells are refractory to clinically relevant concentrations of three different DNA damaging agents.

## Sensitivity of rhabdoid tumor to mithramycin is not due to DNA damage

In an effort to further exclude DNA damage as the mechanism of action of mithramycin in RT cells, we directly compared the concentration of drug required to induce DNA damage to the IC50 of mithramycin and the chemotherapeutic agent etoposide. Again, the RT cells proved to be resistant to the DNA damaging properties of etoposide, requiring between 15 and 20 μM to induce DNA damage as measured by γH2AX staining and Western blot analysis (Fig 2A). Importantly, this genotoxic concentration exactly matches the 15 μM IC50 of etoposide in this cell line, suggesting the importance of DNA double strand breaks to the mechanism of etoposide in RT cells despite the high IC50. Even with these relatively high concentrations of etoposide, there was limited apoptosis as measured by live cell cleaved caspase 3,7 imaging (Fig 2B). These findings are consistent with an intrinsic resistance of the cell to etoposide-induced DNA damage. In contrast, the rhabdoid tumor cells were 1,000 times more sensitive to mithramycin than etoposide with an IC50 of 20 nM. Further, the RT cells required more than 200 nM of mithramycin or roughly 10 times higher exposure to induce even minimal DNA damage (Fig 2C and D). Despite the lack of DNA damage, 100 nM mithramycin triggers an impressive increase in apoptosis as measured by cleaved caspase 3,7 (Fig 2D). To exclude signaling effects as the cause of the phosphorylation of H2AX, we confirmed these results by COMET assay again showing almost no DNA damage at active concentrations of mithramycin in rhabdoid tumor cells (Fig EV1A). These data

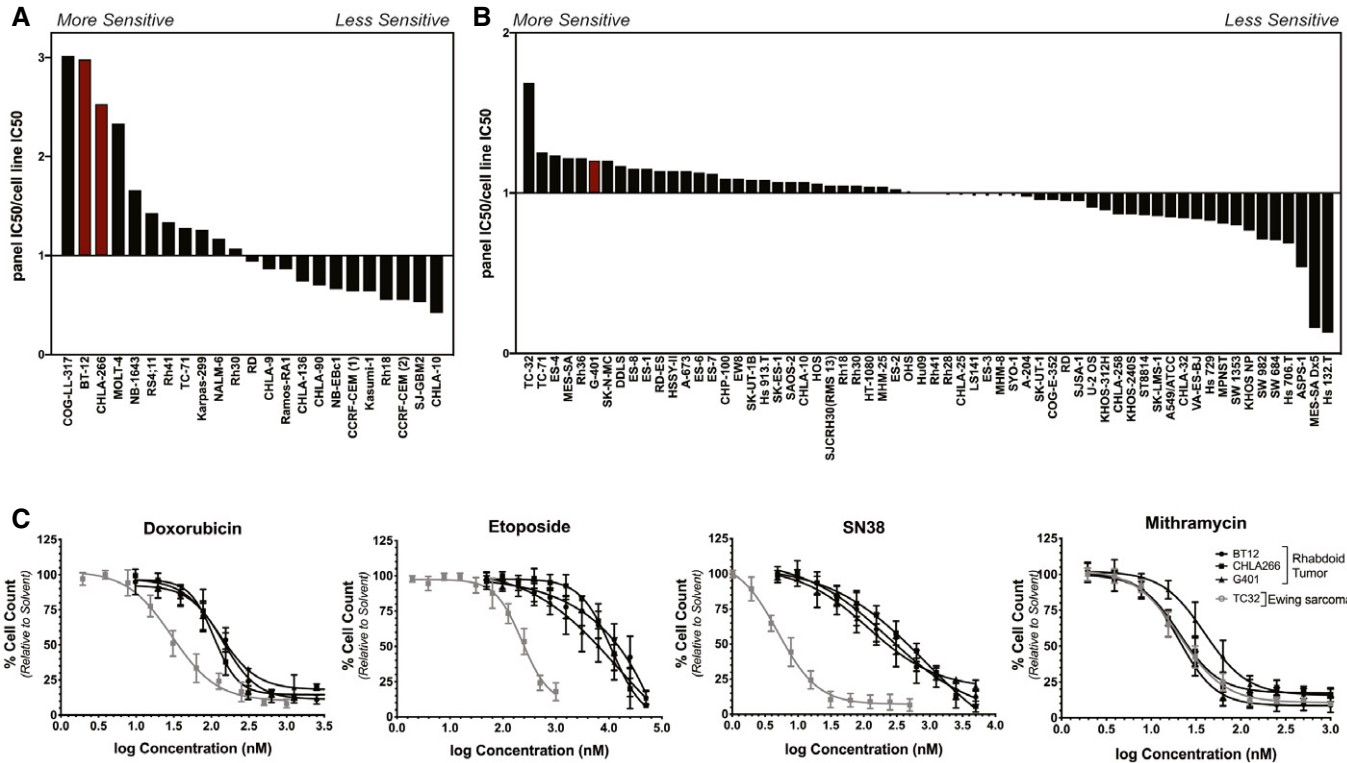

**Figure 1. Rhabdoid tumor is sensitive to mithramycin, but not general chemotherapeutic agents.**

A Graph of the ratio of the median IC50 of the entire panel to that of each cell line generated from a screen of 23 pediatric cancer cell lines. Rhabdoid tumor cell lines (red) cluster toward the left of the graph indicating these cell lines are more sensitive to mithramycin. These results confirm a previously published screen (Osgood *et al*, 2016).

B Graph of the ratio of the median IC50 of the entire panel to that of each cell line generated from a published screen of 445 agents in 62 sarcoma cell lines (Teicher *et al*, 2015). The rhabdoid tumor cell line, G401 (red), appears on the left side of the graph indicating this cell line is more sensitive to mithramycin.

C Dose–response curves of rhabdoid tumor and Ewing sarcoma cell lines. RT cell lines (black) are sensitive to mithramycin treatment with a similar IC50 value as TC32 ES cells (gray). RT cell lines are not sensitive to three broadly active chemotherapeutic agents: etoposide, doxorubicin, or SN38. Data represent mean with standard deviation derived from three independent experiments.

suggest that the DNA damage is a minor component of the mechanism of action for mithramycin in this tumor type. Importantly, at concentrations of mithramycin that approached the IC50, the compound showed a different phenotype of proliferation arrest with some cell cycle changes, some apoptosis and evidence of differentiation with the accumulation of lipid, and the appearance of an eccentric nucleus in a subset of cells in culture (Figs 2E and EV1B–D, Appendix Table S2).

To further exclude the role of DNA damage in the mechanism of action for mithramycin sensitivity, we utilized EC8042, a second-generation analogue of mithramycin that is currently under development for the clinic. Importantly, EC8042 is 20–40 times less toxic than mithramycin in multiple species (Osgood *et al*, 2016). EC8042 required a concentration of 750 nM to induce any appreciable DNA damage in only one of two RT cell lines. In addition, similar to mithramycin, this genotoxic concentration is 10-fold greater than the 75 nM IC50 of the compound (Figs 2F and EV1E). Further, the absence of DNA damage was even more prominent in G401 cells. There was no appreciable increase in γH2AX relative to solvent with drug treatment at concentrations 140 times higher than the IC50 (1,000 vs. 75 nM). Instead, with prolonged

exposure, EC8042 arrested cellular proliferation and induced what appeared to be mesenchymal differentiation with the accumulation of lipid and migration of the nucleus to the cell periphery in a subset of cells, similar to mithramycin (Fig 2E and G). The accumulation of lipid was confirmed by oil red O staining and induction of PPARγ mRNA expression (Fig 2H and I). Together, these data indicate the mechanism of action for mithramycin and EC8042 is not dependent on DNA damage. While DNA damage occurs at relatively high concentrations of mithramycin and EC8042, the observed phenotype of cellular differentiation at sub-IC50 concentrations likely reflects fundamental biology of the disease, particularly with prolonged exposure to the less toxic analogue EC8042.

## Mithramycin inhibits SWI/SNF binding to chromatin to restore H3K27me3

In order to better understand the cellular sensitivity of RT to mithramycin and characterize the differentiation phenotype, we looked at the relationship between SWI/SNF, mithramycin, and H3K27me3. It is known that the biallelic deletion of SMARCB1 establishes a block

in differentiation and favors proliferation by redistributing SWI/SNF in the genome (Alver *et al*, 2017; Wang *et al*, 2017). Since both SWI/SNF and mithramycin bind the minor groove, we hypothesized that mithramycin blocks binding of SWI/SNF to chromatin (Sastry & Patel, 1993). This would be consistent with previously described observations of Quinn, Peterson *et al* that showed a related

compound, chromomycin, competitively inhibits binding of SWI/SNF to DNA (Quinn *et al*, 1996).

Rhabdoid tumor cells were exposed to 100 nM mithramycin, collected, and fractionated into total, cytoplasmic soluble, nuclear soluble, and chromatin-bound fractions (modified from Rothbart *et al*, 2012). BT12 cells showed a loss of binding of three SWI/SNF

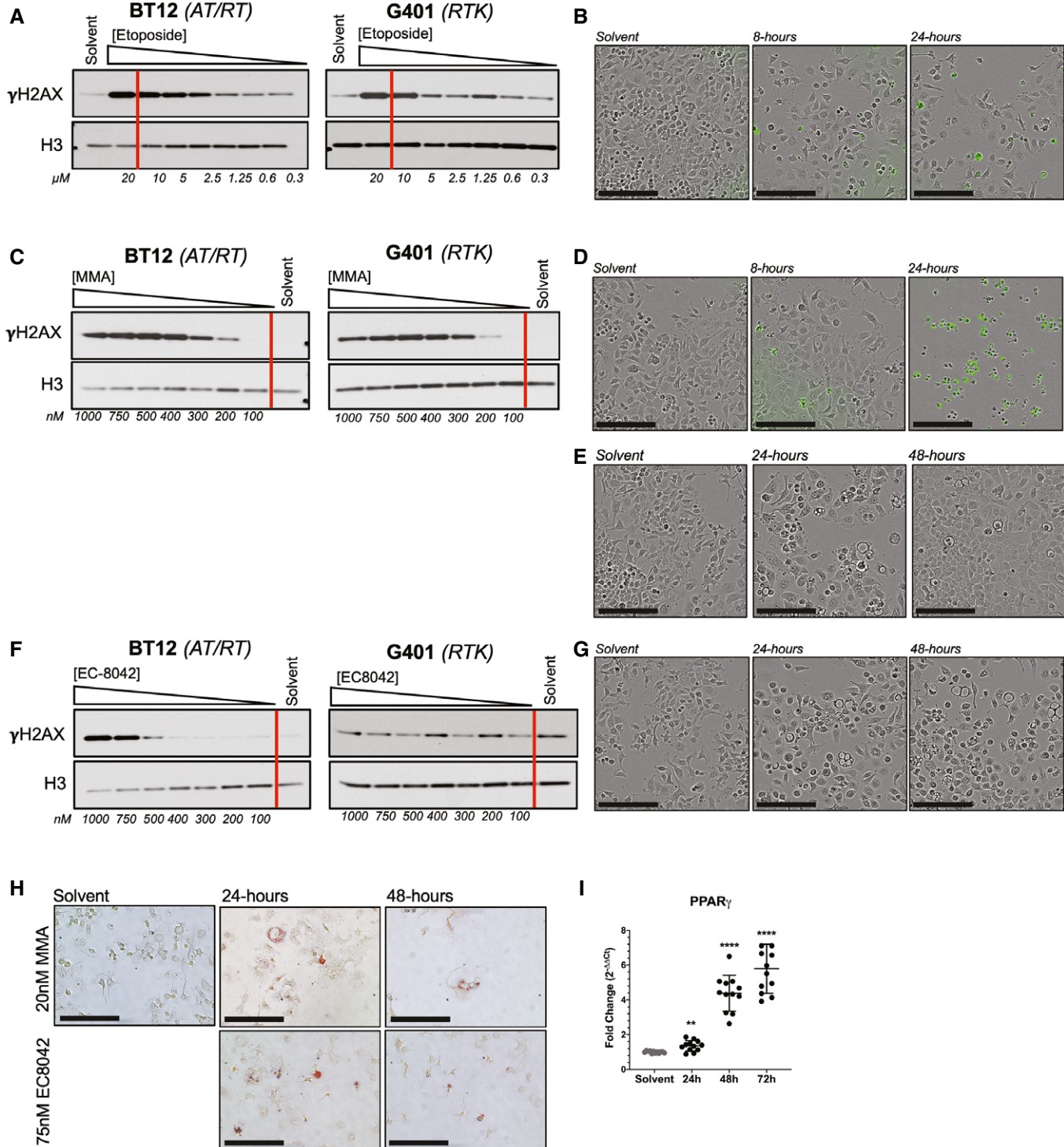

**Figure 2.**

**Figure 2. Mithramycin sensitivity is not due to DNA damage and drives divergent phenotypes in rhabdoid tumor.**

A, B  Western blot showing concentration-dependent increase in γH2AX following 8-h exposure to etoposide in BT12 and G401 RT cells (A). Red bar indicates the measured IC50 (Fig 1C). Despite induction of DNA damage, 15 μM etoposide does not lead to apoptosis as indicated by live cell imaging in the presence of cleaved caspase 3,7 (CC3,7) reagent that fluoresces with caspase activation (B). Scale bar (lower left): 150 μm.

C, D  Western blot showing concentration-dependent increase in γH2AX following 8-h exposure to mithramycin (MMA) in BT12 and G401 RT cells (C). Red bar indicates the measured IC50 (Fig 1C). 100 nM mithramycin induces apoptosis at 8 h as measured by CC3,7 fluorescence without the presence of DNA damage (D). Scale bar (lower left): 150 μm.

E  BT12 cells treated with 20 nM MMA show a different cellular phenotype than with 100 nM. Beyond 24 h of exposure, there is evidence of mesenchymal differentiation and the appearance of maturing adipocytes. Scale bar (lower left): 150 μm.

F, G  Western blot showing concentration-dependent increase in γH2AX following 8-h exposure to EC8042 in BT12 and G401 RT cells. Red bar indicates the measured IC50 (Fig 1C). In contrast to apoptotic induction, 75 nM EC8042 exhibits evidence of mesenchymal differentiation, similar to 20 nM MMA (E,G). Please note that solvent control for (E) and (G) is the same although different fields are shown. Scale bar (lower left): 150 μm.

H  Mesenchymal differentiation confirmed with oil red O staining of lipid deposits following 20 nM MMA or 75 nM EC8042 treatment for 24 and 48 h. Please note that solvent control for 20 nM MMA and 75 nM EC8042 is the same. Scale bar (lower left): 150 μm.

I  PPARγ mRNA expression is induced following 20 nM mithramycin treatment as measured by qPCR fold change relative to GAPDH ($2^{ddCT}$). **$P$ = 0.002, ****$P$ = 0.0001. Data represent mean with standard deviation derived from three independent experiments. $P$-values were determined by one-way ANOVA using Dunnett test for multiple comparisons.

Source data are available online for this figure.

subunits (SMARCC1, BRD9, and SMARCE1) to chromatin at 18 h (Fig 3A). Importantly, the loss of SWI/SNF binding was not seen in U2OS cells (human osteosarcoma cell line) which have SMARCB1 expressed (Fig 3B). It is notable that displacement of BRD9, a subunit in the non-canonical complex, was also observed in U2OS cells, though these cells do not have a dependence on this complex. As additional evidence of displacement, it is known that loss of binding of SWI/SNF subunits to chromatin is associated with rapid degradation by the proteasome (Sohn *et al*, 2007; Kadoch & Crabtree, 2013). Indeed, we showed no change in the transcription of SMARCE1 or SMARCC1 by qPCR in BT12 and G401 cells at the identical time and concentration associated with a loss of expression (Figs 3C and EV1F). Further, the loss of expression was rescued by treatment with the proteasome inhibitor bortezomib in BT12 and G401 cells (Figs 3D and EV1G). We confirmed the loss of binding of SWI/SNF by chromatin immunoprecipitation and quantitative PCR (ChIP-qPCR) at two SWI/SNF binding sites in the genome, *MYT1* and *CCND1*, previously described by others and confirmed in our laboratory (Figs 3E and EV2A) (Boulay *et al*, 2017; Harlow *et al*, 2019).

Since SWI/SNF and PRC2 exist in an antagonistic relationship, we next sought to determine whether the loss of SWI/SNF binding altered this dynamic to restore the accumulation of H3K27me3. Indeed, we showed using chromatin immunoprecipitation that H3K27me3 accumulated at the identical *MYT1* and *CCND1* locus where SWI/SNF was displaced to a similar degree and magnitude (Fig 3F). Importantly, neither the loss of SWI/SNF binding nor accumulation of H3K27me3 was seen at a *GAPDH* locus control in the genome (Fig EV2A and B). This led to a concentration-dependent increase in H3K27me3 that was evident by Western blot and in another model G401 cells (Fig 3G and H). These effects were dependent on deletion of SMARCB1 as complementation using a doxycycline inducible SMARCB1 eliminated the dose dependent accumulation of H3K27me3 seen with mithramycin exposure in G401 cells (Fig 3I).

## The loss of proliferation is dependent on SWI/SNF eviction, H3K27me3 accumulation, and favors SMARCB1 deletion

In order to link mithramycin-mediated eviction of SWI/SNF binding and accumulation of H3K27me3 to the described cellular hypersensitivity of rhabdoid tumor, we next examined the importance of SMARCB1 deletion to these effects. Chromatin fractionation in G401 cells in the presence of 100 nM mithramycin again showed a loss of SMARCB1-deficient SWI/SNF binding to the genome (Fig 4A). Further, complementation of these cells with the same inducible SMARCB1 that eliminated H3K27me3 accumulation (see Fig 3I) led to the inability of mithramycin to compete SMARCE1, a subunit of the canonical SWI/SNF complex, off chromatin consistent with the need for the deletion for this activity (Fig 4B).

Further, the effects on proliferation were dependent on the subsequent accumulation of H3K27me3 and also favor SMARCB1 deletion. BT12 and G401 rhabdoid tumor cells were exposed to 100 nM mithramycin for defined periods of time followed by removal and replacement with drug-free medium. Beyond the identical 8-h time point where SWI/SNF was displaced, there was no recovery of cell proliferation with drug removal (Figs 4C and EV2C). Importantly, this time point was also associated with the first visible increase in H3K27me3 by Western blot (Fig 4D). In order to further link the mechanism of the suppression of cellular proliferation to the accumulation of H3K27me3, we silenced EZH2, the catalytic subunit of PRC2 responsible for H3K27me3, and evaluated the impact on mithramycin sensitivity. Silencing of EZH2 with either siRNA or with the small molecule inhibitor, tazemetostat, in BT12 made the cells less sensitive to the drug shifting the IC50 almost 2-fold from 20 to 37 nM (Figs 4E and EV2D). Conversely, silencing of SMARCB1 in U2OS cells to genocopy SMARCB1-deleted rhabdoid tumor increases the sensitivity of the cells to mithramycin almost 2-fold from 202 to 120 nM (Figs 4F and EV2E). Further, the deletion of SMARCB1 in U2OS cells triggered an increase in H3K37me3 that was not otherwise evident with the SMARCB1 wild-type complex (Fig 4G). These data are consistent with the sensitivity of RT cells to mithramycin being related to SMARCB1-depletion, SWI/SNF displacement, and the accumulation of H3K27me3. Therefore, these features are likely important components to the mechanism of action of mithramycin and may explain the cellular sensitivity of rhabdoid tumor to this drug.

## Mithramycin triggers a change in cellular state and transcriptional switch in RT cells

In an effort to better understand how the loss of proliferation translates into the differentiation phenotype, we next determined how

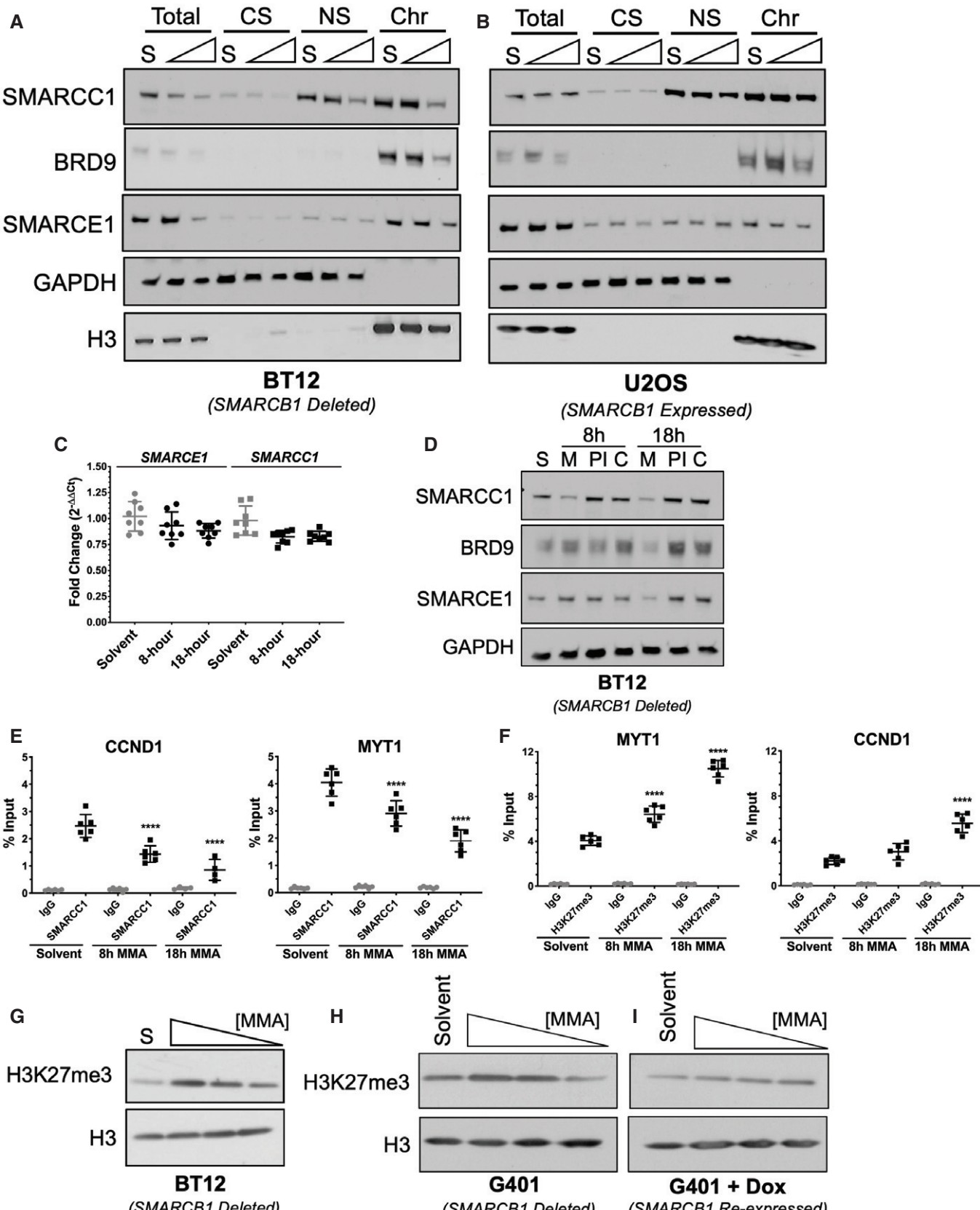

Figure 3.

**Figure 3. Mithramycin induced morphological changes are dependent on SWI/SNF eviction and the induction of H3K27me3.**

A, B  Mithramycin displaces SMARCC1 and SMARCE1 SWI/SNF subunits from chromatin in a time-dependent manner in BT12 rhabdoid tumor (A) but not U20S osteosarcoma (B) cells. Western blot analysis showing whole cell lysate (Total), cytoplasmic soluble (CS), nuclear soluble (NS), and chromatin-bound (Chr) fractions collected after exposure to solvent (S) or 100 nM mithramycin for 8 or 18 h and probed for the SWI/SNF subunits (BRD9, SMARCC1 or SMARCE1) or H3 (chromatin fraction control) and GAPDH (soluble fraction control).

C  SMARCC1 and SMARCE1 mRNA expression does not change following 100 nM mithramycin treatment as measured by qPCR fold change relative to GAPDH ($2^{ddCT}$). Data represent mean with standard deviation derived from three independent experiments.

D  Addition of the proteasome inhibitor (bortezomib) rescues the loss of protein expression following 8- or 18-h mithramycin treatment. BT12 cells were treated for 8 or 18 h with solvent (S), mithramycin (M, 100 nM), bortezomib (PI, 2.5 μM), or combination of 100 nM mithramycin and 2.5 μM bortezomib (C).

E  Loss of SWI/SNF occupancy at defined loci in the genome as measured by chromatin immunoprecipitation qPCR (ChIP-qPCR) at previously described SWI/SNF target genes, MYT1 (8 h, ****$P$ = 0.0001; 18 h, ****$P$ = 0.0001) and CCND1 (8 h, ****$P$ = 0.0001; 18 h, ****$P$ = 0.0001). ChIP quantitation is percent input (ng of immunoprecipitated DNA/input DNA *100) determined by absolute quantitation of sheared chromatin relative to a standard curve. Data represent mean with standard deviation derived from three independent experiments. $P$-values were determined by one-way ANOVA using Dunnett test for multiple comparisons.

F  Chromatin immunoprecipitation qPCR (ChIP-qPCR) of H3K27me3 at MYT1 (8 h, ****$P$ = 0.0001; 18 h, ****$P$ = 0.0001) and CCND1 (8 h, $P$ = 0.02; 18 h, ****$P$ = 0.0001). H3K27me3 occupancy is increased in a time-dependent manner. ChIP quantitation is percent input (ng of immunoprecipitated DNA/input DNA *100) determined by absolute quantitation of sheared chromatin relative to a standard curve. Data represent mean with standard deviation derived from three independent experiments. $P$-values were determined by one-way ANOVA using Dunnett test for multiple comparisons.

G–I  Western blot showing concentration-dependent increase in H3K27me3 following exposure to 100, 50, 25 nM mithramycin for 18 h in BT12 (G) and G401 (H) cells relative to loading control (H3). Re-expression of SMARCB1 following doxycycline treatment inhibits the mithramycin-dependent effects on H3K27me3 amplification (I).

Source data are available online for this figure.

these changes impacted gene expression and chromatin structure. We needed to exclude non-specific mechanisms of suppression, such as general transcription inhibition, global chromatin remodeling, or known targets of the drug such as SP1 as the cause of the loss of proliferation. The goal was to relate the observed phenotypes to changes in gene expression and focal chromatin remodeling of pathways critical to rhabdoid tumor.

Therefore, we performed RNA sequencing, ATAC sequencing, and ChIP sequencing following 8 and 18 h of 100 nM mithramycin treatment to characterize the overall effect on gene expression and chromatin architecture. We chose 8 h because there was no associated DNA damage and no cell death at this concentration and time (see Figs 2C and 4C). In addition, it was the inflection point of cell survival beyond which there was no recovery (see Fig 4C). We evaluated the impact at 18 h to trend the effects over time. Mithramycin did not cause a global suppression of gene expression with only 617 genes (5.4% [$n$ = 11,515], $P$ < 2.8e-6) decreasing in expression by a conservative logFC of 2 at 8 h. In addition, by lowering the stringency of gene expression to a logFC of 1.5 only 10.1% of 11,515 genes were differentially expressed (Fig 5A–C). Strikingly, despite continued exposure to relatively high concentrations of drug, even fewer genes, only 54 (0.5%, $P$ < 1.6e-6) were repressed with drug exposure at 18 h with a logFC cutoff of 2 (Fig 5D–F). Instead, the gene expression changes favored a change from repression to induction consistent with an epigenetic switch leading to the induction of 6–7 times more genes ($n$ = 368 with a logFC > 2 or 3.2%, $P$ < 1.6e-6) with continued drug exposure over time. It is notable that even this induction of gene expression occurred in a small percentage of genes captured by RNA sequencing, thus excluding global changes in gene expression as the causative mechanism of the suppression of cellular proliferation (Appendix Fig S1).

Next, we excluded known targets of the drug as the cause of these cellular phenotypes. SP1 is a well-described target of mithramycin that is clearly repressed by the drug in multiple cell types (Snyder et al, 1991; Remsing et al, 2003). Treatment of RT cells with mithramycin caused a loss of expression of SP1, an effect confirmed by qPCR (Fig 5G and EV3A). Consistent with the described mechanism, this decrease in expression was associated with a loss of

binding of SWI/SNF to the SP1 promoter and a gain of H3K27me3 in the same region (Fig 5H and I). Further, silencing of subunits of SWI/SNF, SMARCC1, and SMARCA4 also caused a loss of expression of SP1 (Fig EV3B). However, siRNA silencing of SP1 did not impact cell viability thus excluding SP1 loss as the cause of the sensitivity of RT cells to the drug (Fig EV3C).

In contrast, pathways known to be important for RT cell survival were modulated by the drug in a manner consistent with the change in gene expression from repression to induction described above. FGSEA analysis clearly demonstrated reversal of 10-differentiation gene signatures pathways previously independently identified as aberrantly activated in primary AT/RT at 8 h of treatment (Fig 5J) (Wang et al, 2017). Further, the drug showed reversal of multiple pro-proliferative oncogenic programs with 8 h of drug exposure including Notch, Hedgehog, Wnt, STAT, and TGFβ (Fig 5K). Importantly, these pathways have been linked to aberrant SMARCB1-deficient SWI/SNF activity in rhabdoid tumor (Chakravadhanula et al, 2015; Johann et al, 2016; Torchia et al, 2016). The FGSEA captured signatures associated with adipogenic differentiation (NES = 2.50, $q$-value = 0.004) as well as apoptosis (NES = 2.18, $q$-value = 0.004), consistent with the cellular phenotypes observed in culture (Figs 2D and E, and 5L). Overall, the RNA-sequencing data excluded general transcriptional effects and instead favored a focal change in chromatin, triggering an epigenetic switch to reverse the oncogenic transcriptome in a manner consistent with the cellular phenotype.

## Focal changes in chromatin structure trigger promoter reprogramming and a differentiation phenotype

Recent studies identified a non-canonical SWI/SNF complex as a synthetic lethal target in RT that binds both promoters and CTCF motifs to drive oncogenic programs (Michel et al, 2018). This model suggests eviction of SWI/SNF would elicit regional focal changes, likely at promoters, as well as a redistribution of H3K27ac to activate multiple downstream targets associated with cellular differentiation. Importantly, this is likely a direct binding interaction between CTCF and SWI/SNF (Marino et al, 2019).

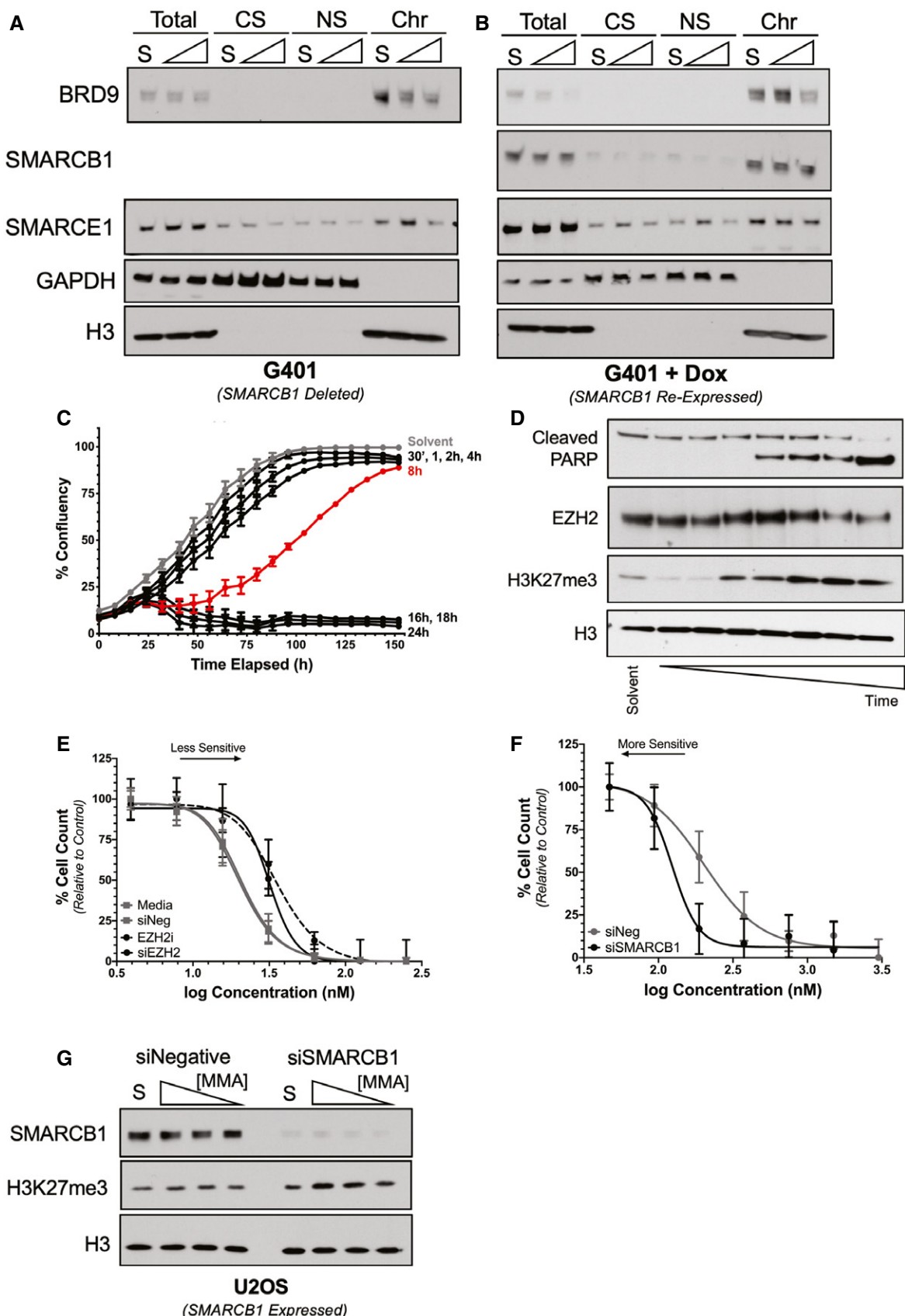

**Figure 4.**

**Figure 4. H3K27me3 amplification and SMARCB1 deletion drives mithramycin sensitivity in rhabdoid tumor.**

A, B   Mithramycin displaces BRD9 and SMARCE1 SWI/SNF subunits from chromatin in a time-dependent manner in G401 (A) but only BRD9 with SMARCB1 re-expression in G401 (B) cells. Western blot analysis showing whole cell lysate (Total), cytoplasmic soluble (CS), nuclear soluble (NS), and chromatin-bound (Chr) fractions collected after exposure to solvent (S) or 100 nM mithramycin for 8 or 18 h and probed for the SWI/SNF subunits (BRD9, SMARCB1 or SMARCE1) or H3 (chromatin fraction control) and GAPDH (soluble fraction control).

C   BT12 cells show a threshold of exposure that leads to irreversible growth inhibition. The cells were exposed to 100 nM MMA for the indicated times followed by replacement with drug-free medium. Beyond 8 h (red) of mithramycin exposure, the cells do not recover proliferative potential and exhibit a phenotype consistent with cell death. Data represent mean with standard deviation derived from three independent experiments.

D   Mithramycin leads to H3K27me3 amplification in a time-dependent manner that precedes the induction of apoptosis as measured by the cleavage of PARP. Western blot lysates collected at 2, 4, 6, 8, 12, 18, and 24 h of continuous 100 nM mithramycin treatment.

E   Suppression of EZH2 expression and activity antagonizes mithramycin activity in BT12 rhabdoid tumor cells. Data represent dose–response curves of mithramycin in BT12 cells following a 48-h exposure in the presence of siRNA silencing of the PRC2 subunit EZH2 or treatment with the EZH2 inhibitor EPZ-6438 relative to mithramycin alone (media) or a non-targeting siRNA (siNeg). Data represent mean with standard deviation derived from three independent experiments.

F   Suppression of SMARCB1 sensitizes U2OS osteosarcoma cells (wild-type SWI/SNF) to mithramycin. Data represent dose–response curves of mithramycin in BT12 cells following a 48-h exposure in the presence of siRNA silencing of the SWI/SNF subunit SMARCB1 relative to a non-targeting siRNA (siNeg). Data represent mean with standard deviation derived from three independent experiments.

G   Western blot showing concentration-dependent increase in H3K27me3 following exposure to 100, 50, 25 nM mithramycin for 18 h in SMARCB1-silenced U2OS cells relative to loading control (H3). Knockdown of SMARCB1 following siRNA suppression triggers mithramycin-dependent H3K27me3 amplification, while siNegative (control) does not have an effect on H3K27me3 following mithramycin exposure.

Source data are available online for this figure.

To interrogate this model, we developed a novel double spike-in approach for the ATAC-seq using lambda phage and phiX to control for library complexity normalization and amplification artifact (Appendix Fig S2). Strikingly, there were relatively few changes in chromatin accessibility genome-wide consistent with the limited number of gene expression changes that occur with drug treatment and excluding global changes in chromatin structure as the causative mechanism of drug sensitivity and activity (Fig 6A, Appendix Fig S3). In addition, motif analysis identified CTCF as the top and highly significant downregulated motif following mithramycin treatment ($P$ = 1e-58; Fig 6B).

In order to link the loss of the CTCF motif to the change in gene expression from 8 to 18 h, we developed a chromHMM model from primary rhabdoid tumors sequenced in TARGET and overlaid the differentially accessible ATAC-seq peaks (Fig 6C). We also included a H3K27ac ChIP-seq to capture the epigenetic switch and induction of gene expression that we observed in the RNA-sequencing data (Fig 5D–F). Consistent with the known biology of RT, the regions that were differentially bound with H3K27ac heavily favored promoters as 75% of the increased H3K27ac DBRs mapped to the promoters, while only 30% of promoters decreased (Fig 6D). Motif analysis of peaks that gain H3K27ac at 18 h captured multiple transcription factors known to be associated with SWI/SNF including ETS, TP53, AP-1, ATF3, FRA1, and FOSL2 (Fig 6D) (Lee *et al*, 2002; Boulay *et al*, 2017; Kelso *et al*, 2017; Vierbuchen *et al*, 2017). As expected, the differentially accessible regions that lose accessibility were relatively non-specific favoring heterochromatin (Appendix Fig S3I). It is likely that these transcription factors contribute to the molecular switch associated with these effects of mithramycin. Importantly, genes identified to have both a reduction in chromatin accessibility and gene expression show reversal of the same pro-survival pathways elucidated by FGSEA including Wnt, Notch, and PI3K-mTOR, as well as 25 different transcription factors, consistent with the effects triggering a change in cellular state (Appendix Table S3).

An analysis of specific loci found alterations in H3K27ac with mithramycin exposure at genomic loci characterized by others as locations for ncSWI/SNF and/or SWI/SNF binding and at sites associated with a block in differentiation in this cell type (*ID3, JUND,*

*CDK6, and CDK2*; Fig 6E and F) (Michel *et al*, 2018; Erkek *et al*, 2019). Importantly, promoter reprogramming captured the differentiation phenotype as *ADIPOR1* and *BMP1*, two critical factors that control adipogenesis and bone morphogenesis, show an increase in promoter accessibility (Fig 6G). It is notable, the peaks in *BMP1* as well as multiple other genes overlapped with published ATAC-seq peaks that were gained following SMARCB1 complementation in rhabdoid tumor cells (Fig EV4) (Weissmiller *et al*, 2019). Overall, these data exclude a general effect on transcription as causative for the cellular sensitivity and instead are consistent with removal of SWI/SNF triggering gene expression changes to drive a specific cellular phenotype.

## Mithramycin shows activity in an intramuscular rhabdoid tumor xenograft

In order to translate our observed *in vitro* mechanism *in vivo*, we treated mice bearing RT xenografts to a continuous exposure of mithramycin administered by a surgically implanted osmotic diffusion pump. We chose an intramuscular xenograft model with G401 cells to exclude limitations on the CNS penetration of the drug and to model the sarcoma variant of the tumor. In addition, we used the osmotic drug elution pump in order to model a lower dose continuous exposure to try to capture the differentiation phenotype. Mice with established tumors of a minimum size of 100 mm$^3$ were treated with 2.4 mg/kg mithramycin given over 3 days with an alzet pump. We compared continuous infusion to intermittent bolus IP injections and did indeed find that the continuous infusion was more effective, repressed growth, and even caused a dramatic regression of a tumor that was 500 mm$^3$ at the start of treatment that persisted for almost 2 weeks after discontinuing treatment. This regression persisted for almost two weeks following discontinuation of treatment. Unfortunately, most tumors progressed following cessation of treatment. Importantly, continuous infusion recapitulated the mechanism of growth suppression described *in vitro*. There was a marked increase in H3K27me3 staining likely responsible for the effects on viability (Fig 7A). Unfortunately, we could not escalate the dose further because of toxicity.

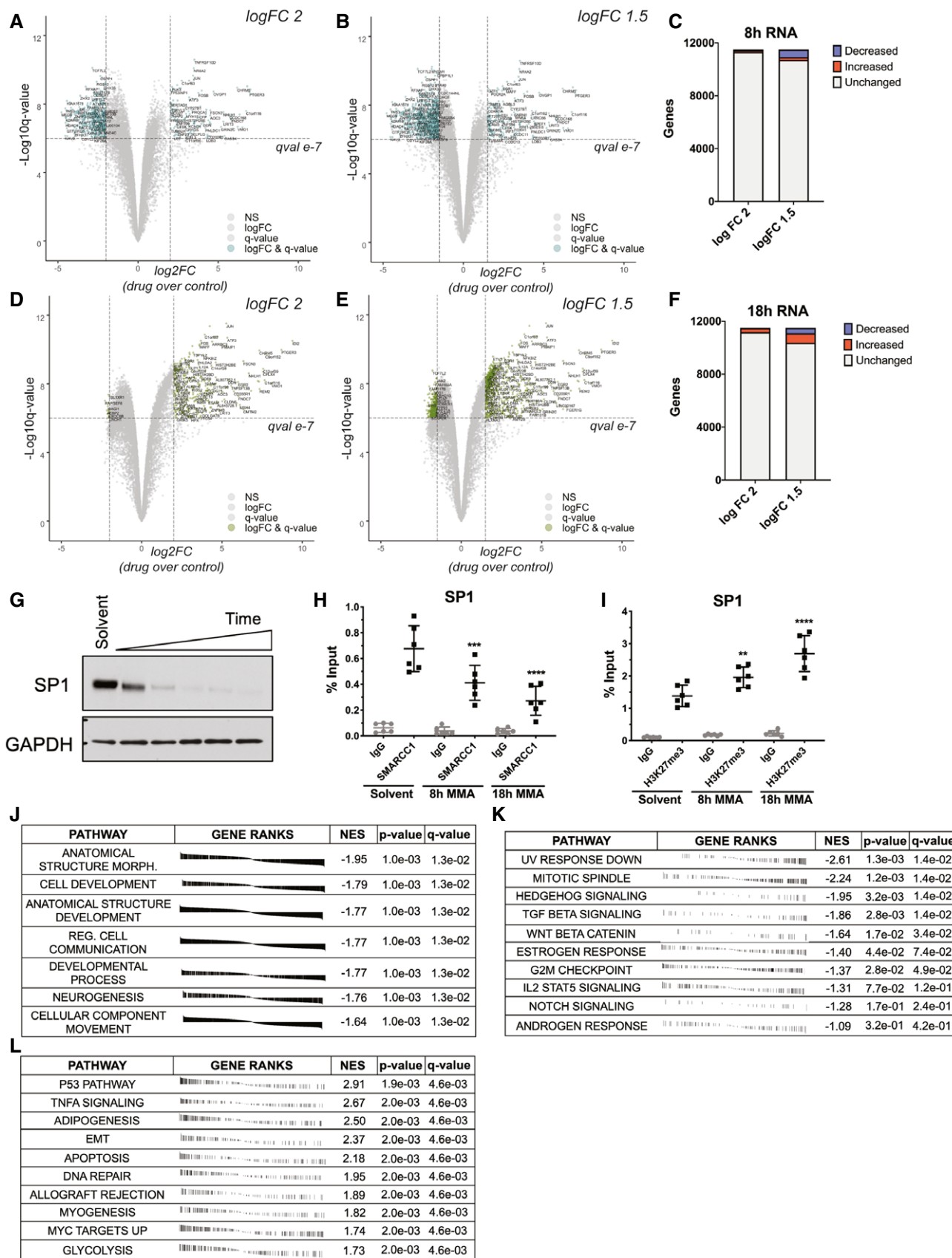

**Figure 5.**

**Figure 5.  Mithramycin does not lead to general transcription inhibition and instead favors pro-survival pathways.**

A–C   Volcano plot showing gene expression trends at 8-h MMA treatment. Dashed lines represent a 2 $\log_2$FC (drug over control) (A) or 1.5 $\log_2$FC (drug over control) (B) and 10e-7 *q*-value threshold. Quantification of induced and repressed genes in (C). Labels indicate names of genes that meet the logFC and *q*-value threshold.

D–F   Volcano plot showing gene expression trends at 18-h MMA treatment. Dashed lines represent a 2 $\log_2$FC (drug over control) (D) or 1.5 $\log_2$FC (drug over control) (E) and 10e-7 *q*-value threshold. Quantification of induced and repressed genes in (F). Labels indicate names of genes that meet the logFC and *q*-value threshold.

G     SP1 protein expression is reduced following mithramycin exposure. Western blot showing suppression of SP1 expression compared to loading control (GAPDH) after 100 nM mithramycin exposure for 1, 4, 8, 12, and 18 h.

H, I   The loss of SP1 expression is associated with a decrease in SWI/SNF occupancy of the SP1 promoter and an increase in H3K27me3. Data represent ChIP-qPCR analysis following 8 or 18 h of 100 nM mithramycin exposure and immunoprecipitation of SMARCC1 (8 h, ***P = 0.0006; 18 h, ****P = 0.0001) (H) and H3K27me3 (8 h, ** P = 0.009; 18 h, ****P = 0.0001) (I). ChIP quantitation is percent input (ng of immunoprecipitated DNA/input DNA *100) determined by absolute quantitation of sheared chromatin relative to a standard curve. Data represent mean with standard deviation derived from three independent experiments. *P*-values were determined by one-way ANOVA using Dunnett test for multiple comparisons.

J     fgsea analysis of gene ontology terms upregulated in primary AT/RT tumors as described in (Wang *et al*, 2017) following 8 h of mithramycin exposure. Mithramycin downregulates the expression of these pathways. *P*-values derived from the fgsea analysis package.

K, L   fgsea analysis of hallmark pathways downregulated after 8 h of MMA exposure (K) or upregulated after 18-h MMA exposure (L). *P*-values derived from the fgsea analysis package.

Source data are available online for this figure.

## EC8042 leads to marked tumor regression and mesenchymal differentiation *in vivo*

In an effort to improve the activity of mithramycin and increase the clinical relevance of the described effects, we evaluated the ability of the second-generation mithramycin analogue EC8042 to recapitulate these effects. EC8042 is a second-generation mithramycin analogue that is more than 10 times less toxic than mithramycin in multiple species but retains a similar IC50 in RT as the parent compound (75 vs. 20 nM) (Osgood *et al*, 2016). This compound is currently being developed for the clinic. Mice with established > 100 mm$^3$ G401 rhabdoid tumor xenografts were treated with 3 days of continuous infusion of EC8042 at a total dose of 30 mg/ kg. All mice experienced striking regressions of their well-established tumors including many with large tumors > 400 mm$^3$ (Fig 7B). Bioluminescent imaging in a subset of mice showed almost no detectable tumor at the end of infusion (Fig EV5B). Retreatment was not possible due to IACUC limitations on additional surgeries to implant another drug eluting pump. However, several mice were cured of their disease with a single 3-day infusion of EC8042 and never showed tumor recurrence 140 days after treatment (Fig 7C). The effects occurred with limited toxicity, and importantly, the total amount of EC8042 administered was six times less than the amount of EC8042 administered in a previous study in Ewing sarcoma (Osgood *et al*, 2016). Mice experienced limited weight loss (average of 19%) that completely resolved with cessation of drug infusion and experienced no electrolyte abnormalities or liver enzyme elevation, which is the known dose limiting toxicity (Fig EV5C, Appendix Table S4).

Importantly, EC8042 drove the combined apoptotic and differentiation phenotype by the described mechanism of action. The tumor showed amplification of H3K27me3 and induction of apoptosis although the apoptosis was relatively minor (Fig 7D). DNA damage was not an important mechanism *in vivo* and was not markedly different than control. Similar to the *in vitro* finding, there was striking evidence of mesenchymal differentiation of the xenograft with the appearance of trabecular-like ossification as well as cartilage and adipocytes (Fig 8). The presence of osteoblasts and embedded osteocytes in the trabecular architecture provides further support for EC8042 inducing osteogenesis (Fig EV5D). Finally, we confirmed calcification of the tumor tissue with micro-computed tomography

(micro-CT; Fig 8). Importantly, the appearance of a mesenchymal differentiation phenotype is supported by rhabdoid tumor cell of origin studies indicating a mesenchymal origin and clinical evidence supporting mesenchymal features in patients although the contribution of the micro-environment in this model can't be excluded (Rorke *et al*, 1996). In addition, the PCA of BT12 cells treated with mithramycin clusters with gene signatures from bone (Fig EV5E). The differentiation phenotype is not fully penetrant and is represented by mixed lineage, likely due to transcriptional and epigenetic heterogeneity in the xenograft as well as influences of the micro-environment. Nevertheless, this phenotype leads to a significant increase in survival of mice bearing the rhabdoid tumor xenografts treated with EC8042.

# Discussion

In this study, we identify mithramycin and the second-generation analogue EC8042 as clinical candidates for RT. We show that mithramycin and EC8042 drive a differentiation phenotype *in vitro* and *in vivo* and link the activity to the defining molecular feature of this tumor, dysregulated SWI/SNF. We show displacement of SMARCB1-deficient SWI/SNF from chromatin and the induction of a cellular response characterized by focal chromatin remodeling to promote a change in the distribution of H3K27ac, favoring promoters to reverse the expression of self-renewal gene expression programs and restore the differentiation program and phenotype. In addition, we link these effects to SMARCB1-deficient SWI/SNF and show that they do not occur with SMARCB1 complementation.

The likely reason for the dependence of these effects on SMARCB1 deletion is the altered affinity and distribution of the complex that occurs with SMARCB1 deletion. It is clear that the impaired opposition with PRC2 favors bivalent promoters and this is overcome with either SMARCB1 complementation or EZH2 inhibitors leading to gene activation and restoration of gene expression at these sites. However, it is also known that SMARCB1 deletion leads to a redistribution of SWI/SNF in the genome, with a higher percentage of ncSWI/SNF, colocalized with SMARCA4, favoring proximal promoters and CTCF sites (Alver *et al*, 2017; Nakayama *et al*, 2017; Michel *et al*, 2018; Erkek *et al*, 2019; Wang *et al*, 2019). We clearly show that exposure of cells to mithramycin remodels chromatin at

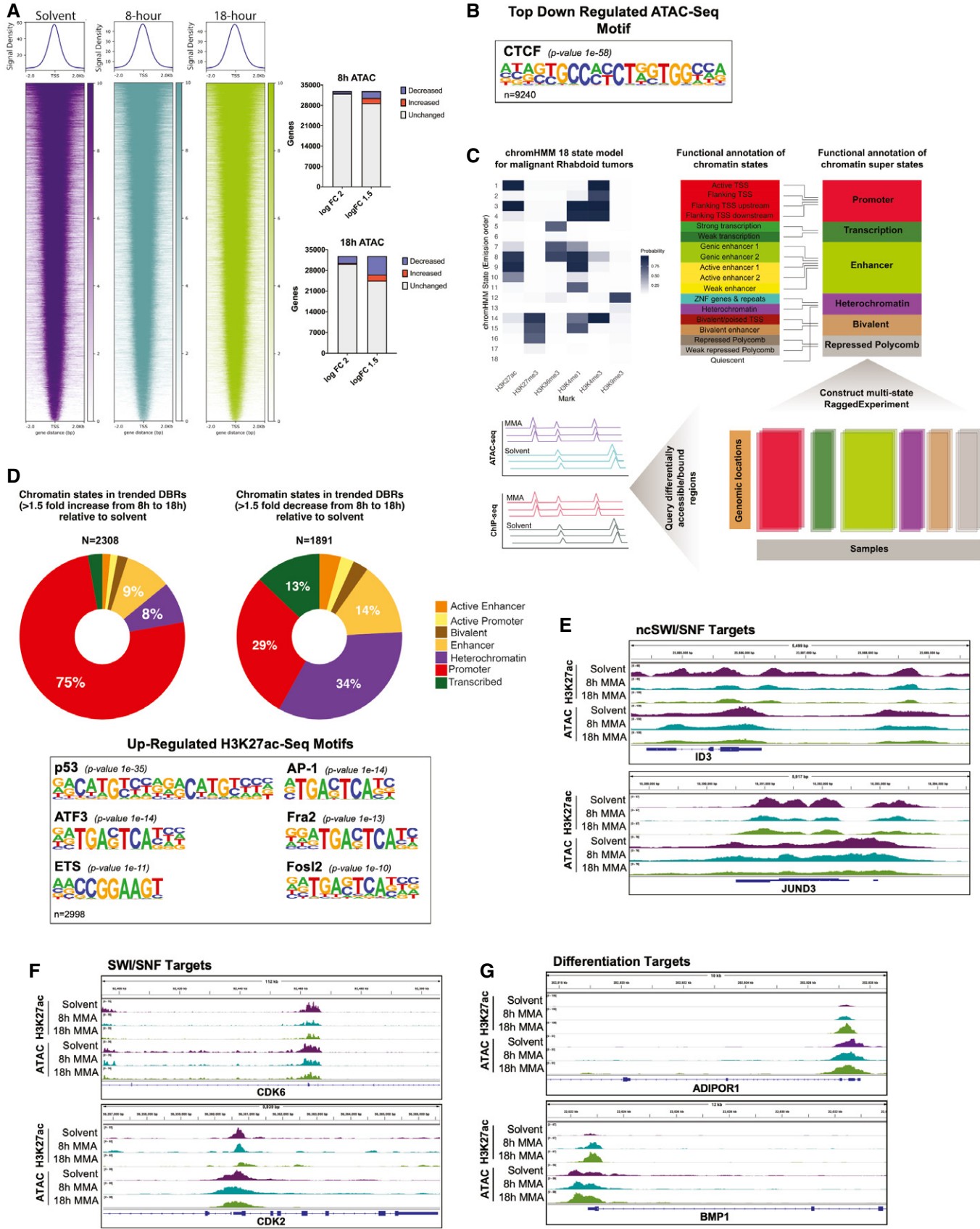

**Figure 6.**

**Figure 6. Mithramycin reprograms rhabdoid tumor promoters to trigger a change in cellular state and favor a differentiation phenotype.**

A   Heatmaps depicting ATAC-seq global chromatin structure following 8 h (middle) and 18 h (right) 100 nM MMA treatment. A 2 kb window is centered on the TSS. Chromatin accessibility clusters at the TSS and does not change globally relative to solvent. Quantification of genes that gain accessibility or reduce accessibility on the right.

B   CTCF is the top downregulated motif from the ATAC-seq analysis (P = 1e-58).

C   Schematic for the 18 state chromHMM model built for MRTs and collapsed into six super states. Chromatin states were called if the state was present in at least 50% of samples. ATAC-seq and H3K27ac ChIP-seq peaks were queried against the six super states. P-values derived from the homer motif analysis package.

D   Donut plots representing the percentage of each chromatin super state across treatment time (from 8 to 18 h) that increased 1.5-fold (left) or decreased 1.5-fold (right). Below, Motif analysis of the top upregulated motifs from the H3K27ac ChIP-seq gene lists that pass a 1.5 logFC. P-values derived from the homer motif analysis package using a binomial algorithm.

E   IGV tracks of rhabdoid tumor genes previously identified to be occupied by non-canonical SWI/SNF (Michel et al, 2018). ID3 and JUND3 decrease in H3K27ac occupancy and chromatin accessibility following exposure to mithramycin for 8 or 18 h.

F   IGV tracks of rhabdoid tumor genes previously identified to gain H3K27me3 upon SWI/SNF loss (Erkek et al, 2019). CDK6 and CDK2 decrease in H3K27ac occupancy and chromatin accessibility following exposure to 8- and 18-h mithramycin treatment.

G   IGV tracks of genes identified to have an increase in accessibility, H3K27ac and gene expression following mithramycin exposure. BMP1 and ADIPOR1 play crucial roles in bone and adipogenic differentiation, respectively.

these sites concomitant with an increase in H3K27me3. It is likely that the lower affinity of SMARCB1-deficient complexes for chromatin allows for this displacement with the drug at these sites altering accessibility to reverse the oncogenic phenotype. Indeed, the most highly enriched motif in our analysis of chromatin accessibility was CTCF (P = 1e-58) and 75% of the chromatin state changes observed demonstrated increased accessibility of promoters. Further, it is likely that higher concentrations of drug, known to be toxic, lead to more widespread effects thus accounting for the differential sensitivity of rhabdoid tumor cells relative to other cell types, including normal cells. It is tempting to speculate that the effect of mithramycin-mediated displacement of SWI/SNF can be generalized to the 20% of all human cancer characterized by mutated or dysregulated SWI/SNF, though more research is needed to explore this hypothesis. However, consistent with this idea, Ewing sarcoma is exquisitely sensitive to this drug and known to dysregulate SWI/SNF by altering the distribution of SWI/SNF through a different mechanism (Boulay et al, 2017).

These effects clearly trigger an epigenetic switch leading to the loss of pro-proliferative gene expression programs described by others and restoration of the differentiation program. The inciting event of this change in cellular state and differentiation phenotype is an increase in H3K27me3 that triggers targeted epigenomic remodeling and provides the link between the observed cellular sensitivity and SWI/SNF. The PRC2-SWI/SNF axis is an important therapeutic target in this tumor and the basis for the use of EZH2 inhibitors in rhabdoid tumor treatment (Knutson et al, 2013). In this report, we provide a complementary approach to the targeting of this axis. We demonstrate that the activity of EC8042 and mithramycin in RT is dependent on this increase in H3K27me3

and can be antagonized by inhibiting EZH2 expression or activity. Interestingly, the overall effect at 18 h of drug exposure is widespread gene activation. This excludes non-specific transcriptional repression as the mechanism of action of the drug. In addition, we show activation of adipogenesis and myogenesis, mesenchymal differentiation signatures (P = 0.002). These gene signatures clearly reflect the cellular phenotype that was observed both in vitro and in vivo. Further, the robust induction of genes associated with the DNA damage response likely accounts for the lack of DNA damage at active concentrations of the drug excluding these non-specific mechanisms as the cause of the cellular hypersensitivity. This relationship with the DNA damage response also may explain the intrinsic resistance of rhabdoid tumor cells to traditional DNA damaging agents.

This study also highlights the importance of mechanistic pharmacology as an approach to improve the clinical translation of targeted agents. We perform a thorough investigation of dose and schedule of administration as a means to achieve the defined mechanism of action for this drug in RT xenografts. Our data show a strong schedule dependence of these tumors for both mithramycin and EC8042. Both compounds are more effective as a continuous infusion, and only the continuous infusion of EC8042 induces durable responses, including complete cures. This schedule recapitulates the described mechanism of action in vivo leading to cellular differentiation and increased H3K27me3 without increasing DNA double strand breaks or γH2AX staining of the tissue. Importantly, these observations provide a pharmacodynamic biomarker of activity in ex vivo and in vivo settings. The induction of H3K27me3 staining of the FFPE tumor tissue was primarily found in tumors that responded to the mithramycin

**Figure 7. EC8042 leads to durable tumor regression in a rhabdoid tumor xenograft model.**

A   Immunohistochemistry analysis recapitulates the biochemistry described in vitro for mithramycin. G401 tumor sections at 20X magnification stained with H&E, cleaved caspase 3 (CC3; apoptosis), or H3K27me3. A marked increase in CC3 that correlates with H3K27me3 staining is seen only in mice treated with the continuous infusion schedule but not vehicle. Scale bar (lower left): 50 μm.

B   Prolonged durable response and cure of mice bearing G401 xenografts treated with 30 mg/kg EC8042 administered continuously over 72 h. Treatment duration indicated by gray shaded box. Asterisk indicates an animal sacrificed due to unknown causes not related to tumor progression or drug toxicity (see text).

C   Kaplan–Meier survival curves indicating extended survival for mice bearing established G401 xenografts treated with the 3-day continuous infusions of EC8042 in (B). The shaded box indicates the duration of treatment. Asterisk indicates an animal sacrificed due to unknown causes not related to tumor progression or drug toxicity (see text).

D   20X image of section of G401 treated tumors stained for H3K27me3, cleaved caspase 3, and γH2AX. The sections compare vehicle to treatment started on day 1 with 30 mg/kg of EC8042 administered continuously for 72 h (3-day pump). H3K27me3 increases and correlates with apoptosis (CC3); however, induction of CC3 is modest. γH2AX staining does not increase with treatment indicating DNA damage is not responsible for these effects. Scale bar (lower left): 50 μm.

analogue EC8042. Importantly, compounds that induce a differentiation phenotype have been found to be highly effective in the clinic in other tumor types with notable examples being arsenic and ATRA for APL, trabectedin for myxoid liposarcoma, and retinoids for neuroblastoma (Flynn *et al*, 1983; Sidell *et al*, 1983;

Forni *et al*, 2009). Consistent with these observations, we show a dramatic suppression of tumor growth of every mouse in the cohort with a single 3-day infusion of the EC8042.

Finally, this study provides important insight into the mechanism of mithramycin. Mithramycin was originally identified as

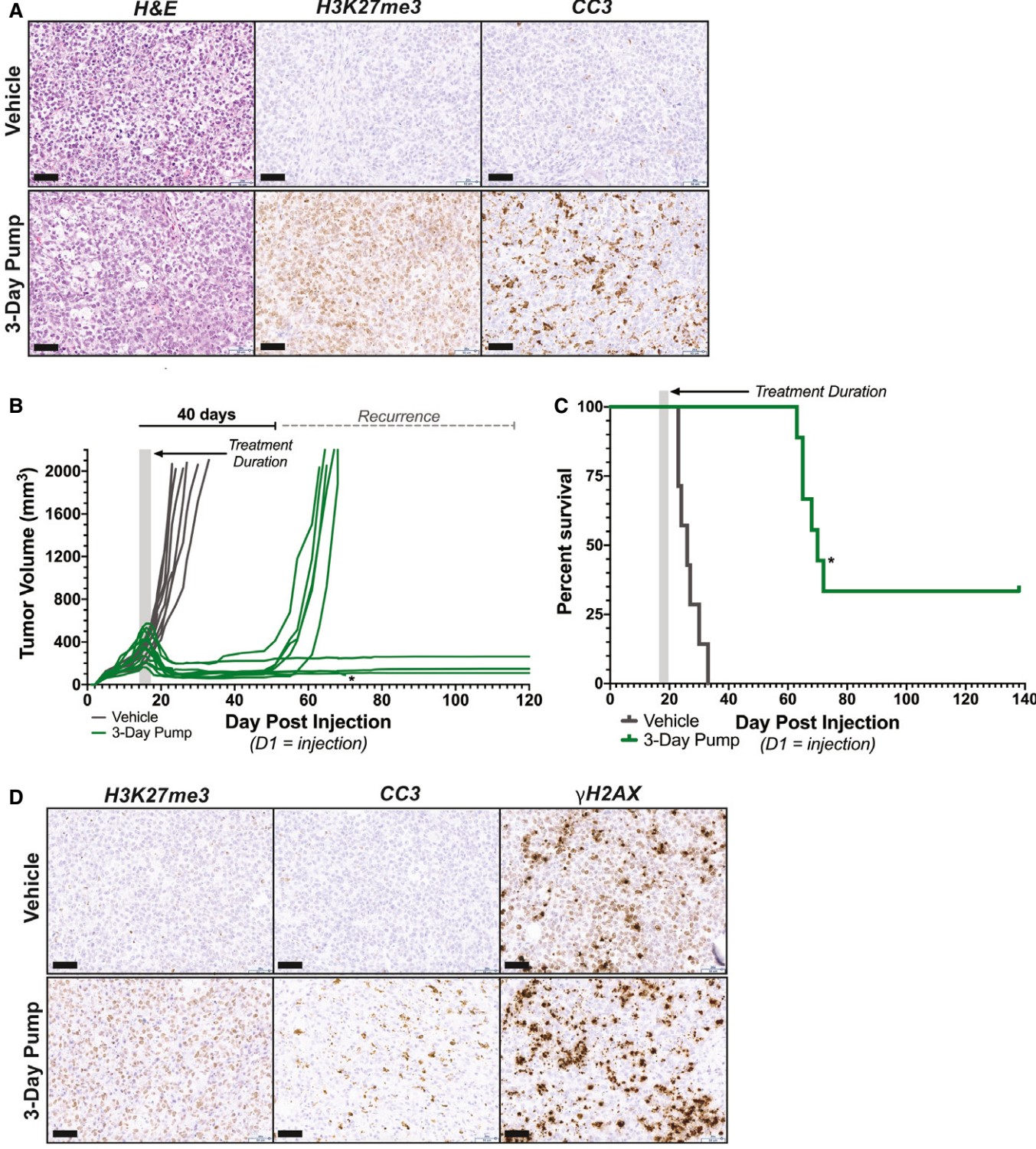

**Figure 7.**

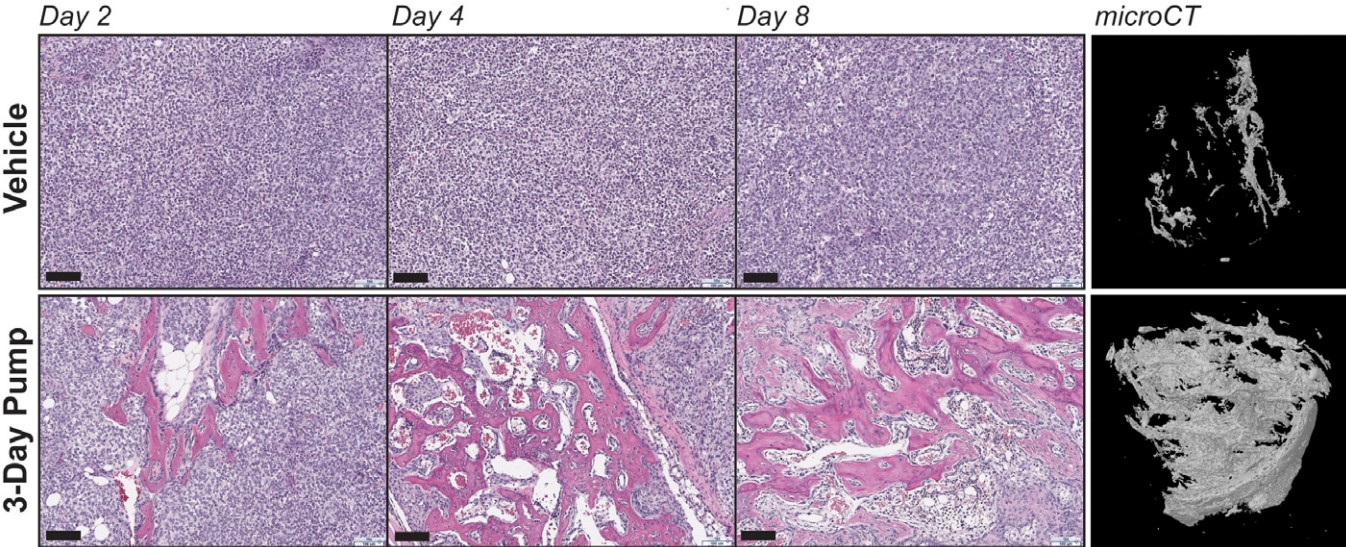

**Figure 8. EC8042 induces mesenchymal differentiation in rhabdoid tumor xenografts.**

Immunohistochemistry analysis of H&E stains from G401 xenograft tumors on 1, 3, and 7 days after treatment with vehicle or 3-day EC8042 infusion. EC8042-treated xenograft tumors exhibit evidence of mesenchymal differentiation compared to vehicle. micro-CT analysis of xenograft tumors on 7 days after treatment exhibit enhanced calcification compared to vehicle. IHC scale bar (lower left): 100 μm.

an anti-cancer agent in the 1950s. While it showed some activity in the clinic, it fell out of favor due to its narrow toxicity profile. It has always been referred to as an SP1 inhibitor with other data suggesting activity against ETS transcription factors and EWS-FLI1, the oncogenic driver of Ewing sarcoma (Grohar *et al*, 2011). The data in this study support why the drug is such a potent SP1 inhibitor and further support a functional relationship between SP1 and SWI/SNF. Mithramycin has been shown to work downstream of SP1 to competitively inhibit SP1 binding to DNA (Snyder *et al*, 1991; Remsing *et al*, 2003). In this study, we show that this compound also works upstream of SP1 in RT, displacing SWI/SNF to drive an increase in H3K27me3 and decrease of H3K27ac at the SP1 promoter, leading to a loss of SP1 expression. Moreover, it is possible mithramycin also disrupts SP1 activity as SWI/SNF interaction with SP1 is required for SP1-dependent gene expression at promoters (Kadam *et al*, 2000; Kadam & Emerson, 2003). However, the suppression of SP1 activity is likely not an important mechanism of the drug in rhabdoid tumor as silencing of SP1 had no effect on cellular proliferation. Nevertheless, it is likely that tumors dependent on SP1 would be particularly sensitive to this combined upstream and downstream targeting. Further, recognition of these mechanisms should aid in the identification of analogues more selective for SP1 vs. SWI/SNF vs. EWS-FLI1. In addition, in tumors dependent on SP1, it provides insight into approaches to develop novel combination therapies that perhaps amplify the targeting of SP1, as has been described with tolfenamic acid in pancreatic cancer (Jia *et al*, 2010).

In summary, this study provides important insight into the biology and therapeutic targeting of SWI/SNF in rhabdoid tumor by mithramycin and EC8042 treatment *in vitro* and *in vivo*. We show that RT cells are hypersensitive to mithramycin and link the activity to the fundamental genetic lesion of the tumor, SMARCB1 deletion, and consequential aberrant SWI/SNF activity. We thoroughly explore the therapeutic development of this compound from a mechanistic and translational perspective thus providing insight into the targeting of this complex as well as a dose, schedule, and biomarker of target inhibition that is clinically translatable. Most importantly, we identify the less toxic second-generation mithramycin analogue EC8042 as a promising novel molecularly targeted agent for patients with rhabdoid tumor.

# Materials and Methods

### Cell culture

BT12 and CHLA266 cell lines were obtained from the Children's Oncology Group repository (www.CCcells.org) and G401 and A204 cells from ATCC. TC32 cells from Dr. Lee Helman (Children's Hospital of Los Angeles) and U2OS cells from Dr. Chand Khanna (Ethos Veterinary Health LLC). TTC642 from Dr. Bernard Weissman (UNC) and doxycycline inducible G401 were a gift from Dr. Charles Roberts (St. Jude Children's Hospital). G402 and BT16 were received from the St. Jude Children's Research Hospital Childhood Solid Tumor Network. Cell lines were routinely monitored for pathogens and cultured as described (Harlow *et al*, 2019).

### Quantitative real-time PCR (RT–qPCR)

BT12 cells were incubated with mithramycin, and the RNA was collected, reverse-transcribed, and PCR amplified as previously described (Harlow *et al*, 2019). Expression was calculated with standard ddCT methods relative to GAPDH and solvent controls. PCR primer sequences are available in Appendix Table S5.

## Cell proliferation assays

IC50s were determined by non-linear regression in duplicate for three independent experiments using MTS assay CellTiter96 and confirmed with real-time proliferation assays on the Incucyte Zoom after 48 h of treatment (Harlow *et al*, 2019). Cytotoxicity relative to a panel of other pediatric cancer cell lines was determined at 96 h by the Pediatric Preclinical Testing Program as described (Kang *et al*, 2011).

## Western blot

BT12 cells were incubated with mithramycin and lysed with 4% lithium dodecyl sulfate (LDS) buffer. Fractionation lysates were collected with cytoplasmic lysis buffer (25 mM HEPES pH 8.0, 50 mM KCl, and 0.5% NP-40) and the nuclei pellet lysed with 4% LDS buffer. Thirty microgram of protein was resolved and transferred as described (Harlow *et al*, 2019). The membranes were blocked with 5% milk in TBS-T and probed with Abcam (H3K27me3, GAPDH), EMD Millipore (SP1), and Cell Signaling (Cleaved PARP, BRD9, SMARCB1, SMARCC1, SMARCE1, γH2AX, H3) antibodies. Antibody information is available in Appendix Table S5.

## Chromatin fractionation

Three million BT12, G401, or U2OS were incubated with 100 nM mithramycin or PBS control for 8 or 18 h, washed and collected in PBS, and fractionated as previously described (Harlow *et al*, 2019) with minor adjustments. Initially, the nuclei are isolated with 320 mM sucrose, 8 mM Tris–HCl (pH 7.5), 4 mM MgCl$_2$, and 0.8% Triton X. Nuclei were released with dounce homogenization, and the cytoplasmic fraction was isolated with centrifugation of the sample at 1,500 *g* for 5 min at 4°C. The fractionation of the nuclear soluble and chromatin-bound fractions was performed as previously described. Antibody information is available in Appendix Table S5.

## RNA sequencing

RNA was extracted and submitted for $1 \times 75$ bp sequencing. Libraries were prepared from 500ng of total RNA as described (Harlow *et al*, 2016). Reads were aligned to hg19 using STAR (v2.7.0f) (Dobin *et al*, 2013). The index was prepared using default parameters except for --sjdbOverhang 75 and --sjdbGTFfile, where the Gencode v19 annotations were used. Default parameters were used for alignment with the following modifications: --readFilesCommand zcat --outReadsUnmapped None --quantMode GeneCounts --outSAMtype BAM SortedByCoordinate. Gene-level transcript quantification was performed using STAR's built-in quantification algorithm as noted in the modified alignment parameters. Libraries were normalized using trimmed mean of *M*-values (TMM) after filtering for low abundance transcripts using the R (v3.6.1) package edgeR (Robinson *et al*, 2010; Robinson & Oshlack, 2010). Differential expression analysis was carried out using limma-voom (v3.40.6) (Law *et al*, 2014; Ritchie *et al*, 2015). Significant genes were determined using a cutoff of $q < 0.05$ (Benjamini–Hochberg). Heatmaps were generated using the pheatmap (v1.0.12) package. GSEA was performed with the functional gsea (v1.10.1) package (preprint:

Sergushichev, 2016). Reactome pathway analysis was performed with the reactomePA (v1.28.0) package (Yu & He, 2016). Primary explant normal skull osteoblast raw RNA-seq data were downloaded from the SRA study SRP038863 (Rojas-Pena *et al*, 2014). Data were aligned and preprocessed as described above. PCA was carried out using $\log_2(\text{TPM} + 1)$ counts and the prcomp function from the stats package (v3.6.1). Results were plotted using ggplot2 (v3.2.1) and the viridis (v0.5.1) packages. Sample and library variability was calculated using quantro (v1.22.0) (Hicks & Irizarry, 2015).

## Chromatin Immunoprecipitation with quantitative PCR

BT12 cells were incubated with 100 nM mithramycin or PBS control for 8 or 18 h. Cells were cross-linked, lysed, and sheared as described (Harlow *et al*, 2019). 10 µg solubilized chromatin was immunoprecipitated with 1 µg mouse IgG and 1 µg H3K27me3 (Abcam); 2 µg rabbit IgG and 2 µg SMARCC1 (Cell Signaling); 1 µg rabbit IgG; and 1 µg H3K27ac (Active Motif). Antibody–chromatin complexes were immunoprecipitated and purified as described (Harlow *et al*, 2019). ChIP DNA was quantified with SYBR green relative to a standard curve generated with chromatin from the respective sample for each primer set. qPCR as described above was performed with the following primer sets (GAPDH, MYT1, SOX2, CCND1, SP1). Antibody information and PCR primer sequences are available in Appendix Table S5.

## Chromatin Immunoprecipitation with high throughput sequencing (ChIP-seq)

BT12 cells were incubated with 100 nM mithramycin or PBS control for 8 or 18 h. Cells were cross-linked, lysed, and sheared as described (Harlow *et al*, 2019). 10 µg solubilized chromatin was immunoprecipitated with 1 µg rabbit IgG and 1 µg H3K27ac (Active Motif). Antibody–chromatin complexes were immunoprecipitated and purified as described (Harlow *et al*, 2019). Libraries for Input and IP samples were prepared from 10 ng of input material and either 10 ng or all available IP as described (Harlow *et al*, 2019). Antibody information is available in Appendix Table S5.

## ChIP-seq analysis

Reads were aligned using bwa mem (v0.7.17), duplicate marked with samblaster (v0.1.24), and filtered and converted to BAM format using samtools (v1.9) (Li *et al*, 2009; Faust & Hall, 2014). BAMs were ingested into R (v3.6.1) and processed using csaw (v1.18.0). A 150 bp sliding window with a 50 bp step size was used to summarize the read counts with a maximum fragment size set to 800 bp (Lun & Smyth, 2014; Lun & Smyth, 2016). Background was estimated using a 5 kb sliding window where reads were binned and summarized, and known hg19 blacklist regions were excluded (Amemiya *et al*, 2019). Regions having signal greater than $\log_2(3)$ fold change over background were retained for differential analysis. The first principal component was regressed out of the data due to a batch effect being present prior to downstream analysis. Differential analysis was carried out using csaw and edgeR, fitting a quasi-likelihood (QL) negative binomial generalized log-linear model that estimates the prior QL dispersion distribution robustly. Differentially bound regions (DBRs) were generated using an ANOVA-like test or

individual contrasts. *P*-values were combined across clustered sites using Simes' method to control the cluster false discovery rate as implemented in the combineTests function in csaw. Clustered DBRs were defined as having a *q* < 0.05 (Benjamini–Hochberg). Heatmaps were generated with deeptools (v3.4.3).

### Assay for transposase accessible chromatin with high throughput sequencing (ATAC-seq)

BT12 cells were treated with 100 nM mithramycin for 8 or 18 h or PBS control for 18 h. 25,000 cells were used to perform omni-ATAC with minor modifications (Corces *et al*, 2017). Prior to transposition, 0.1 ng lambda phage (Thermo Fisher Scientific) was added to the nuclei pellet. Transposition was carried out for 60 min at 37°C. After purification, 0.1 ng phiX DNA were added to the transposition DNA prior library amplification. Libraries were amplified and purified as described.

Finished libraries were size-selected to retain fragments between 200–800 bp using double sided SPRI selection with Kapa Pure Beads. Indexed libraries were pooled, and 75 bp, paired end sequencing was performed on an Illumina NextSeq 500 sequencer using a 150 bp HO sequencing kit (v2). Base calling was done by Illumina NextSeq Control Software (NCS) v2.0, and output of NCS was demultiplexed and converted to FastQ format with Illumina Bcl2fastq2 v2.20.0.

### ATAC-seq analysis

Reads were aligned to hg19 as described above for ChIP-seq, and the enterobacteria phage lambda genome (NC_001416.1) was added as additional contigs. BAMs were processed using csaw (v1.18.0) in R (v3.6.1). A 150bp sliding window with a 50 bp step size was used to summarize the read counts with a maximum fragment size set to 500 bp. The background was estimated using a 1 kb sliding window where reads were binned and summarized. Regions having signal greater than $\log_2(3)$ fold change over background were retained for differential analysis. Libraries were normalized using RUVg from the RUVSeq (v1.18.0) package, with the lambda reads as the control "genes" (Risso *et al*, 2014). PCA plots examining the effects of increasing k were plotted using EDASeq (v2.18.0) (Risso *et al*, 2011). Differentially accessible regions (DAR) were computed by fitting a similar model as described for ChIP-seq, but with the RUV weights added. Statistical significance was determined as *q* < 0.05 (Benjamini–Hochberg). Library complexity analysis was performed using preseqR (v4.0.0) (Deng *et al*, 2015). Chromatin conformation was inferred by extending methods as previously described and implemented in compartmap (v1.3) (Fortin & Hansen, 2015). Filtered read counts are summarized within a bin, pairwise Pearson correlations are computed across samples within a group, and the first principle component describes the chromatin conformation state using the sign of the eigenvalue. Chromosome-wide compartment dissimilarity scores were computed relative to solvent by calculating 1—Pearson correlations. Heatmaps were generated with deeptools (v3.4.3).

### chromHMM analysis

ChIP-seq data were downloaded from phs000470 for 19 MRT patients and aligned to the hg38 assembly with bwa mem,

duplicates marked and removed with samblaster, and converted to BAM format with samtools (Chun *et al*, 2016). The following chromatin marks were used as input to construct the chromHMM (v1.18) model: H3K4me1, H3K4me3, H3K9me3, H3K27Ac, H3K27me3, and H3K36me3 (Ernst & Kellis, 2012; Ernst & Kellis, 2017). Additionally, matched input samples were used for local thresholding during the binarization step. BAMs were binarized using default values and segmented using the Roadmap 18-state core model collapsed into 6 "super states" (Roadmap Epigenomics *et al*, 2015). The original and collapsed chromatin state calls were combined into a RaggedExperiment (v1.8.0) object. Differential regions were queried for overlaps to specific states, where a consensus state was called as having at least 50% of patient samples having the same inferred state from chromHMM. Donut plots were generated using significant (*q* < 0.05) trended changes in accessibility (ATAC-seq) or binding (ChIP-seq) from 8 to 18 h of mithramycin treatment relative to solvent.

### siRNA knockdown

RNAiMax Lipofectamine was added to siRNA targeting SMARCA4, SMARCB1, SMARCC1, EZH2, or SP1 and allowed to complex. BT12 cells were added to the mixture and incubated for 30 h (SP1) or 48 h (SMARCA4 and SMARCC1) before collection for qRT–PCR analysis. BT12 or U2OS cells were added to the mixture and incubated for 30 h (SP1) or 48 h (EZH2 and SMARCB1) before collection for MTS, qPCR, or Western blot analysis. Antibody information and PCR primer sequences are available in Appendix Table S5.

### COMET assay

BT12 rhabdoid tumor cells were treated with vehicle, 100 nM mithramycin, or 15 μM etoposide for 8 h and then harvested in PBS. The comet assay was performed according to manufacturer's protocol (Abcam ab238544) with the exception that the cells were stained with 1× SYBR Green (Thermo Fisher Cat. S7563).

### Luciferase cells

CMV-luciferase plasmid was linearized with HF-Sal1 and transfected into G401 cells as described (Grohar *et al*, 2011). Cells were expanded under G418 selection and confirmed to be pathogen-free.

### Xenograft experiments

$5 \times 10^6$ G401-luc cells were injected intramuscularly into the gastrocnemius of 8- to 10-week-old female homozygous nude mice (Crl; Nu-*Foxn1*$^{Nu}$; Van Andel Institute, Grand Rapids MI). Tumors were established to a minimum of 100 mm$^3$, and tumor volume was measured daily by caliper and the volume determined as described (Harlow *et al*, 2019). Mice (*n* = 12 per cohort) were treated with mithramycin given at 1mg/kg mithramycin IP (8 mg/kg total), 2.4 mg/kg in a 3-day continuous infusion alzet micro-osmotic pump (model 1003D) or vehicle (PBS supplemented with magnesium or calcium) per the identical schedules (*n* = 12). EC8042-treated mice were treated with 30 mg/kg in a 3-day infusion. All experiments were performed in accordance with and the approval of the Van Andel Institute Institutional Animal Care and Use Committee

### The paper explained

**Problem**

There is a tremendous need for novel therapeutic approaches for rhabdoid tumor and the more than 20% of human cancers characterized by dysregulation of the SWI/SNF chromatin remodeling complex. Rhabdoid tumor is a pediatric cancer that is most commonly characterized by biallelic deletion in SMARCB1 of the SWI/SNF chromatin remodeling complex. This mutation creates a dependence of this cancer on residual SWI/SNF activity. While promising therapeutic approaches targeting associated complexes known to be influenced by dysregulated SWI/SNF, such as PRC2, are being evaluated in the clinic, the direct therapeutic targeting of oncogenic SWI/SNF has not been explored.

**Results**

In this report, we identify EC8042 as an inhibitor of SMARCB1-deficient SWI/SNF and explore the therapeutic development of this compound from a mechanistic and translational perspective. Inhibition of SMARCB1-deficient SWI/SNF triggers an epigenetic switch to cause differentiation of rhabdoid tumor mouse xenografts into bone. This switch results in complete cures of three of eight mice treated with a single 3-day infusion of EC8042.

**Impact**

This study provides insight into the therapeutic targeting of the SWI/SNF chromatin remodeling complex and highlights EC8042 as a promising therapeutic candidate for rhabdoid tumor. In addition, the study provides a pharmacodynamic marker of target blockade as well as an optimized dose and schedule of administration to aid in the translation of this compound to the clinic for this tumor.

(IACUC). Mice were group housed in micro-isolator cages on a 12-h light/dark cycle with *ad libitum* access to water and rodent chow (Lab Diet 5021, Purina Mills, Richmond, IN). Investigators were not blinded to the treatment groups.

### Bioluminescence imaging

Mice were administered Firefly D-Luciferin with intraperitoneal injections. After injection, anesthesia was administered throughout the image acquisition (3% isoflurane at 1 l/min $O_2$ flow). Bioluminescent images were taken 10 min after injection using the AMI-1000 imaging system.

### Micro-computed tomography (micro-CT)

Mineralized tissue within tumors was examined using the SkyScan 1172 micro-computed tomography system (Bruker MicroCT). Tumors were scanned in 70% ethanol using an X-ray voltage of 60 kV, current of 167 μA, and 0.5 mm aluminum filter. The pixel resolution was set to 2,000 × 1,200, with an image pixel size of 8 μm. A rotation step of 0.40 degrees and 360° scanning was used. 2D cross-sectional images were reconstructed using NRecon 1.7.4.6. A volume of interest (VOI) was defined for each tumor using Data-Viewer 1.5.6.3. A region of interest (ROI) around the tumor was defined, and 3D files were generated using CTAn 1.18.8.0. Representative 3D images were created using CTvol 2.3.2.0.

### Tissue staining and immunohistochemistry

Tissues were decalcified in 10% EDTA (pH 8.0) and sectioned following paraffin embedding as described (Harlow *et al*, 2019). Tissue was incubated with H3K27me3 (Abcam, 1:250), cleaved caspase 3 (Cell Signaling, 1:250), or γH2AX (Abcam, 1:800) washed and then secondary antibody (Envision + System HRP labeled polymer Anti-Rabbit, Dako 1:100).

### Project statistics

qPCR data are normalized to solvent (mRNA expression data) or input (ChIP data) as fold change from three independent experiments performed in technical duplicate. The *P*-values were determined by one-way ANOVA using Dunnett test for multiple comparisons. Twelve animals per cohort were used assuming the smallest reduction in tumor growth to be 60% based on published data and 80% power to detect a similar reduction between two treatments. Animals were only excluded if tumors were not detected and randomized to ensure an equal distribution of tumor size and attributes among cohorts. Investigators were not blinded to the treatment groups.

## Data availability

The sequencing data have been deposited on GEO under the accession number GSE137404.

**Expanded View** for this article is available online.

## Acknowledgements

Research reported in this manuscript was supported by the National Cancer Institute of the National Institutes of Health under the award number F31-CA236300-01 (M.H.C.) and internal funds from the Van Andel Research Institute (T.J.T. and P.J.G.) and Children's Hospital of Philadelphia (P.J.G.). The COG/ALSF Childhood Cancer Repository is supported by Alex's Lemonade Stand Foundation. The authors would like to thank the Van Andel Genomics, Bioinformatics and Biostatistics, and Pathology and Biorepository Cores for providing next generation sequencing facilities, formal analysis and immunohistochemistry analysis. The authors would like to thank the University of Pennsylvania Flow Cytometry and Genomic Analysis Cores for flow analysis and cell line authentication. The authors would like to thank EntreChem for the use of EC8042. Finally, the manuscript is in honor of J.S. and family.

## Author contributions

MHC and PJG conceptualized the study. MHC, BKJ, EAB, TJT, BOW, and PJG involved in methodology. MHC, EAB, KMS, JER, MHK, CPR, and SMK-G involved in investigation. MHC, BKJ, MHK, CPR, LH, IB, ZBM, GEF, BOW, TJT, and PJG formally analyzed the study. MHC, BKJ, and PJG involved in writing. MHC, TJT, and PJG acquired the funding. PJG involved in resources. BOW, TJT, and PJG supervised the study.

## Conflict of interest

The authors declare that they have no conflict of interest.

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
