## [Review Process File · EMBO Molecular Medicine]

Mithramycin induces promoter reprogramming and differentiation of rhabdoid tumor

Maggie Chasse, Benjamin Johnson, Elissa Boguslawski, Katie Sorensen, Jessica Rosien, Min Kang, Patrick Reynolds, Lyong Heo, Zach Madaj, Ian Beddows, Gabrielle Foxa, Susan Goosen, Bart Williams, Timothy Triche, and Patrick Grohar

DOI: 10.15252/emmm.202012640

Corresponding authors: Patrick Grohar (groharp@email.chop.edu)

Review Timeline:

Submission Date:	30th Apr 20
Editorial Decision:	19th May 20
Revision Received:	20th Oct 20
Editorial Decision:	4th Nov 20
Revision Received:	16th Nov 20
Editorial Decision:	18th Nov 20
Revision Received:	18th Nov 20
Accepted:	20th Nov 20

Editor: Lise Roth

Transaction Report:

19th May 2020

Dear Dr. Grohar,

Thank you for submitting your work to EMBO Molecular Medicine. We have now received feedback from the three reviewers who agreed to evaluate your manuscript. As you will see from the reports below, they appreciate the potential translational interest of the study, but also underline the lack of robust mechanistic analysis.

We will therefore welcome a revised version of your manuscript. Addressing the reviewers' concerns in full will be necessary for further considering the manuscript in our journal, and acceptance of the manuscript will entail a second round of review. EMBO Molecular Medicine encourages a single round of revision only, and acceptance or rejection of the manuscript will depend on the completeness of your responses included in the next, final version of the manuscript. For this reason, and to save you from any frustrations in the end, I would strongly advise against returning an incomplete revision.

We understand that the Covid-19 pandemic affects scientific work at every level. Please contact us if you need more than six months to revise your manuscript.

To submit your manuscript, please follow this link:

Link Not Available

When submitting your revised manuscript, please carefully review the instructions that follow below. Failure to include requested items will delay the evaluation of your revision:

- 1) A .docx formatted version of the manuscript text (including legends for main figures, EV figures and tables). Please make sure that the changes are highlighted to be clearly visible.
- 2) Individual production quality figure files as .eps, .tif, .jpg (one file per figure).
- 3) A .docx formatted letter INCLUDING the reviewers' reports and your detailed point-by-point responses to their comments. As part of the EMBO Press transparent editorial process, the point-by-point response is part of the Review Process File (RPF), which will be published alongside your paper.
- 4) A complete author checklist, which you can download from our author guidelines (<https://www.embopress.org/page/journal/17574684/authorguide#submissionofrevisions>). Please insert information in the checklist that is also reflected in the manuscript. The completed author checklist will also be part of the RPF.

6) Before submitting your revision, primary datasets produced in this study need to be deposited in an appropriate public database (see <https://www.embopress.org/page/journal/17574684/authorguide#dataavailability>). Please remember to provide a reviewer password if the datasets are not yet public. The accession numbers and database should be listed in a formal "Data Availability" section (placed after Materials & Method). Please note that the Data Availability Section is restricted to new primary data that are part of this study.

7) We would also encourage you to include the source data for figure panels that show essential data. Numerical data should be provided as individual .xls or .csv files (including a tab describing the data). For blots or microscopy, uncropped images should be submitted (using a zip archive if multiple images need to be supplied for one panel). Additional information on source data and instruction on how to label the files are available at .

8) Our journal encourages inclusion of *data citations in the reference list* to directly cite datasets that were re-used and obtained from public databases. Data citations in the article text are distinct from normal bibliographical citations and should directly link to the database records from which the data can be accessed. In the main text, data citations are formatted as follows: "Data ref: Smith et al, 2001" or "Data ref: NCBI Sequence Read Archive PRJNA342805, 2017". In the Reference list, data citations must be labeled with "[DATASET]". A data reference must provide the database name, accession number/identifiers and a resolvable link to the landing page from which the data can be accessed at the end of the reference. Further instructions are available at .

9) We replaced Supplementary Information with Expanded View (EV) Figures and Tables that are collapsible/expandable online. A maximum of 5 EV Figures can be typeset. EV Figures should be cited as 'Figure EV1, Figure EV2' etc... in the text and their respective legends should be included in the main text after the legends of regular figures.

- Additional Tables/Datasets should be labeled and referred to as Table EV1, Dataset EV1, etc. Legends have to be provided in a separate tab in case of .xls files. Alternatively, the legend can be supplied as a separate text file (README) and zipped together with the Table/Dataset file. See detailed instructions here:

10) The paper explained: EMBO Molecular Medicine articles are accompanied by a summary of the articles to emphasize the major findings in the paper and their medical implications for the non-specialist reader. Please provide a draft summary of your article highlighting

- the medical issue you are addressing,
- the results obtained and

- their clinical impact.

11) For more information: There is space at the end of each article to list relevant web links for further consultation by our readers. Could you identify some relevant ones and provide such information as well? Some examples are patient associations, relevant databases, OMIM/proteins/genes links, author's websites, etc...

12) Every published paper now includes a 'Synopsis' to further enhance discoverability. Synopses are displayed on the journal webpage and are freely accessible to all readers. They include a short stand first (maximum of 300 characters, including space) as well as 2-5 one-sentences bullet points that summarizes the paper. Please write the bullet points to summarize the key NEW findings. They should be designed to be complementary to the abstract - i.e. not repeat the same text. We encourage inclusion of key acronyms and quantitative information (maximum of 30 words / bullet point). Please use the passive voice. Please attach these in a separate file or send them by email, we will incorporate them accordingly.

Please also suggest a striking image or visual abstract to illustrate your article. If you do please provide a png file 550 px-wide x 400-px high.

13) As part of the EMBO Publications transparent editorial process initiative (see our Editorial at <http://embomolmed.embopress.org/content/2/9/329>), EMBO Molecular Medicine will publish online a Review Process File (RPF) to accompany accepted manuscripts.

In the event of acceptance, this file will be published in conjunction with your paper and will include the anonymous referee reports, your point-by-point response and all pertinent correspondence relating to the manuscript. Let us know whether you agree with the publication of the RPF and as here, if you want to remove or not any figures from it prior to publication.

I look forward to receiving your revised manuscript.

Yours sincerely,

Lise Roth

Lise Roth, PhD
Editor
EMBO Molecular Medicine

Photos 400-800 DPI

*Additional important information regarding figures and illustrations can be found at <http://bit.ly/EMBOPressFigurePreparationGuideline>

***** Reviewer's comments *****

Referee #1 (Comments on Novelty/Model System for Author):

cell lines reflective of human disease.

Referee #1 (Remarks for Author):

The study by Chasse et al. revealed that mithramycin, an antibiotic with anticancer properties, and its less toxic analogue EC8042 exhibit selective antineoplastic effects against Rhabdoid tumor. The authors attributed the anticancer effects of mithramycin to its role in displacing aberrant SWI/SNF complex, the oncogenic driver of the disease, from chromatin and subsequent deposition of H3K27me3 leading to focal promoter reprogramming and driving a combination of apoptosis and differentiation. Overall, the manuscript is well written, the experiments are well-designed and the data sufficiently support the derived conclusions. Also, this work is of clinical relevance since it established EC8042 as a candidate for treating rhabdoid tumors and provided a rationale for testing this drug in 20% of human cancers where one or more subunits of the SWI/SNF complex are mutated.

Minor Comments:

1. The manuscript could benefit from additional DNA damage assays other than γ -H2AX (maybe Comet assays) to accurately assess DNA damage at low concentrations of mithramycin
2. Figure 1B- indicate more explicitly on the waterfall plot the more sensitive and less sensitive cell lines
3. The authors indicated that mithramycin induces cellular differentiation based on morphological changes. Additional evidence is needed to confirm differentiation (maybe by looking at specific markers or using Oil Red O staining to assess differentiation into adipocytes). Figure 2

differentiation- indicate what dose of drug

4. Figure 3A. Mithra leads to loss of SMARCC1 from chromatin but there does not seem to be an increase in soluble fraction. No increase in G401 cells either. Where are the proteins going?

5. 3E need to show action of MITHRA on U2OS control cells.

6. In figure 4, the authors need to show immunoblot confirming EZH2 knockdown. Also, EZH2 inhibitors can be used to confirm the protective effect against mithramycin.

7. RNA seq - did total RNA change per cell indicating global changes in RNA production- were spike in controls used for the RNA-Seq. Figure 5 legend DEF is labeled as 8 hours- I think you mean 18 hours.

Referee #2 (Remarks for Author):

This is an interesting and provocative manuscript addressing an important biological and medical problem. However, the manuscript would benefit from the following improvements:

- 1) Experimental results of the screen in Fig 1A and B are unclear. Please show IC50 or AUC values.
- 2) Phenotypic effects of EC8042 are intriguing. A more detailed evaluation is warranted: what are the effects of sub-IC50 concentrations on cell cycle kinetics and differentiation as measured by flow cytometry?
- 3) Is the apparent reduction in SMARCC1 and SMARCE1 chromatin association dependent on SMARCB1? In other words, what happens to the association of BAF subunits with chromatin as a function of EC8042 treatment upon re-expression of SMARCB1 in rhabdoid tumor cells. This will test the prediction that EC8042 regulates BAF complex assembly in SMARCB1-dependent manner in rhabdoid tumor cells, as opposed to secondary consequences of the drug effects that otherwise are independent of the dysregulated functions of the mutant BAF complexes in tumor cells.
- 4) Similarly, the apparent changes in H3K27 methylation upon EC8042 treatment can be related to the BAF complex by measuring this effect upon re-expression of SMARCB1 that has been found to affect Polycomb antagonism in these tumors.
- 5) What is known about the BAF regulation of SP1 expression in rhabdoid or other cell types? The manuscript should include a discussion whether this has been shown or not. Could SP1 gene expression changes be BAF-independent?
- 6) RNA interference experiments need to include multiple independent siRNA constructs against EZH2, or alternatively genetic complementation rescue with EZH2 cDNA, given the recognized risk of off-target effects of RNA interference.
- 7) Given the proposed model, it would be interesting to examine the effects of EC8042 in the presence of tazemetostat, which has been shown to regulate the BAF-PRC axis in rhabdoid tumors. If direct experiments cannot be done, then a discussion should be included, given that tazemetostat has recently been approved for patients.
- 8) The manuscript favors a model in which EC8042 regulates the BAF-PRC axis: do gene expression measurements support this model, eg does EC8042 exhibit significant changes in PRC gene sets? Are observed gene expression changes consistent with the morphologic mesenchymal

differentiation that is observed?

9) Likewise, do EC8042-induced changes in chromatin accessibility occur at loci regulated by PRC or BAF?

10_ The xenograft effects are impressive. What plasma drug concentrations are achieved in this model, and how do these concentrations compare to the concentrations that induced chromatin and differentiation changes in vitro? Can this analysis inform the types of pharmacokinetic properties that an effective drug should have in human clinical trials?

11) The manuscript would benefit from a discussion of specific mechanisms of action of EC8042, mithramycin and related aureolic acids, with respect to their molecular target(s), their selectivity for these targets, the potential for off-targets, and their relationship with the effects on tumor versus normal cells, particularly in developing tissues. This is a complex and confusing subject, and the new results reported in the manuscript, combined with the provocative model that is proposed, would benefit from this discussion.

12) This statement "The deletion or inactivation of SMARCB1 does not affect SWI/SNF structural integrity." is not supported by the published results. Please see papers by Kadoch et al and Roberts et al that show that the BAF complexes are structurally disrupted by loss of SMARCB1. Likewise, gain-of-function activities have been reported for mutant BAF complexes in synovial sarcoma cells. Residual complexes in rhabdoid tumor cells exhibit mostly loss-of-function phenotypes, eg loss of chromatin association. Please clarify the evidence that the SMARCB1-deficient residual BAF complexes associate with and regulate the expression of new genes or loci?

13) Fig 5 axis is unclear: log fold change in which direction: drug/control or control/drug, please clarify

Referee #3 (Remarks for Author):

This manuscript examines whether mithramycin, a putative SP1 transcription factor inhibitor, will prove effective as an inhibitor of growth of rhabdoid tumor cell lines. The authors have previously shown that mithramycin inhibits the growth of Ewing's sarcoma cell lines. Both types of tumors share defects in the SWI/SNF chromatin remodeling complex. Ewing's sarcoma expresses a fusion protein that knocks the SMARCB1 subunit out of the complex resulting in an abnormal oncogenic complex while rhabdoid tumors have lost SMARCB1 expression leading to a loss of 2 of the 3 normal complexes. In this report, the authors demonstrate that 3 out of 4 rhabdoid cell lines show sensitivity to therapeutically-relevant doses of mithramycin both in culture and in xenografts. They also demonstrate that the mechanism of inhibition does not involve the induction of DNA damage. Thus, mithramycin may provide an effective treatment for patients with rhabdoid tumors, a cancer with a generally dismal prognosis, with the potential for rapid translation into clinical trials. However, most of the other experiments in the manuscript raise serious concerns about the authors' interpretation of the results. Therefore, the mechanisms by which mithramycin inhibits the growth of rhabdoid cell lines remains unclear. The authors should address the concerns below in order establish how mithramycin inhibits the growth of rhabdoid tumors.

Major Comments:

1) Figure 1B- Apparently not all rhabdoid cell lines respond to mithramycin because the A204

rhabdoid cell line did not respond. This cell line was initially listed as an eRMS but was shown to be a rhabdoid tumor cell line in 2002 (PMID: 23882450). The authors should comment on this differential sensitivity. In addition, at least 10 MRT cell lines exist. Why do the authors only present mithramycin sensitivity data for 3 of them?

2) The X axis labelling in Figure 1C is confusing. If this is log concentrations, why does the axis say nM?

3) The authors should provide references for MYT1 and CCND1 as "well-established SWI/SNF binding sites". Only a limited number of groups have found CCND1 as a direct target of the SWI/SNF complex. RNA-seq data has found few, if any, consensus targets for rhabdoid tumors, let alone SWI/SNF-mutant tumors.

4) If mithramycin competes with the SWI/SNF complex for binding to the minor groove of DNA, why wouldn't it evict SMARCC1 and SMARCE1 from chromatin in the U2OS cell line in Figure 3? Are the authors proposing that only the ncBAF complex binds to the minor groove of DNA? If so, then mithramycin should still evict BRD9 and GLTSCR1/GLTSCR1L1 from chromatin in the U2OS cell line, an easy experiment to perform. Otherwise, the results in Figure 3 do not make sense.

5) The authors provide no protein validation for any of their gene expression data, even for the knockdown experiments. This raises serious concerns about the validity of their conclusions.

6) Figure 4- Multiple groups have shown that MRT cell lines are sensitive to EZH2 knockdown or inhibitors due to the restoration of p16INK4A expression. Surprisingly, these authors observe no effect on growth of their MRT cell lines after knockdown of EZH2. How do they explain this discrepancy with previous studies? It was also unclear why cells treated with mithramycin for 16 hours would become irreversibly committed to growth arrest. Removal of the drug should restore the ability of the SWI/SNF complex to bind to chromatin and evict polycomb complexes. Therefore, it seems like mithramycin effects should be easily reversed after 16 hour exposure.

7) If mithramycin suppresses expression of SP1, why doesn't it show up in either the volcano plots in Figure 5 or in the list of genes in Table S1?

8) The ATAC-seq and H3K27ac ChIP-seq data in Figure 5 and in the genome tracks throughout the manuscript are different than other published reports. Why do the authors observe an uneven ATAC-seq signals around TSSs? Normally the signal is symmetrical around the sites (see the Pan et al. PMC6755913 for examples). In addition, H3K27ac usually appears as a dual peak at enhancers and TSSs. Some of the tracks show this common pattern but the heat maps and most of the tracks show a single wide peak. The fact that the authors do not list the program they used to plot the ChIP/ATAC-seq heatmaps makes it difficult to understand these differences.

9) The authors refer to the ncBAF complex that remains in the rhabdoid tumors showing a "gain-of-function activity". In the Discussion, they refer to an "oncogenic SWI/SNF" in rhabdoid tumors as opposed to the "wild-type complex in U2OS cells". This is an incorrect interpretation of the papers from the Kadoch laboratory. The ncBAF complex normally appears in all cells. Because it does not possess SMARCB1 as a subunit, SMARCB1-deficient rhabdoid tumors lose the BAF and PBAF complexes but retain the ncBAF complex. Thus, the absence of the other 2 SWI/SNF complexes allows ncBAF to localize to sites normally occupied by BAF and PBAF, not a gain of function or mutation in the ncBAF members.

Minor Comments:

The authors should stain the cells in Figure 2 E & G with Oil Red to demonstrate lipid deposits.

Referee #1 (Remarks for Author):

The study by Chasse et al. revealed that mithramycin, an antibiotic with anticancer properties, and its less toxic analogue EC8042 exhibit selective antineoplastic effects against Rhabdoid tumor. The authors attributed the anticancer effects of mithramycin to its role in displacing aberrant SWI/SNF complex, the oncogenic driver of the disease, from chromatin and subsequent deposition of H3K27me3 leading to focal promoter reprogramming and driving a combination of apoptosis and differentiation. Overall, the manuscript is well written, the experiments are well-designed and the data sufficiently support the derived conclusions. Also, this work is of clinical relevance since it established EC8042 as a candidate for treating rhabdoid tumors and provided a rationale for testing this drug in 20% of human cancers where one or more subunits of the SWI/SNF complex are mutated.

Minor Comments:

1. The manuscript could benefit from additional DNA damage assays other than γ -H2AX (maybe Comet assays) to accurately assess DNA damage at low concentrations of mithramycin

Thank you. We have included a COMET assay in this revision see Figure EV1 . Again, no DNA damage was found even at 100 nM mithramycin after 8-hour exposure which is higher than the IC50. Etoposide was included as a positive control.

2. Figure 1B- indicate more explicitly on the waterfall plot the more sensitive and less sensitive cell lines

Thank you. We have included this notation and agree that it makes the figure clearer.

3. The authors indicated that mithramycin induces cellular differentiation based on morphological changes. Additional evidence is needed to confirm differentiation (maybe by looking at specific markers or using Oil Red O staining to assess differentiation into adipocytes). Figure 2 differentiation- indicate what dose of drug

In this revision, we have included Oil Red O and qPCR to show induction of PPARG a marker of adipogenic differentiation. These are Figures 2 H, I. Importantly PPARG is induced 6-fold.

4. Figure 3A. Mithra leads to loss of SMARCC1 from chromatin but there does not seem to be an increase in soluble fraction. No increase in G401 cells either. Where are the proteins going?

It is known that removal of SWI/SNF subunits from chromatin leads to degradation of subunits (Kadoch & Crabtree, 2013, Sohn, Lee et al., 2007). In the first version, we assumed this was happening based on the published literature. However, we agree that we should have demonstrated that this was degradation and not simply loss of expression. So, in this revision, we show that there is no change in production of SMARCC1 or SMARCE1 with drug treatment by qPCR in two rhabdoid tumor cell lines. In addition, we show that the subunits are indeed degraded when displaced from chromatin and that the loss of expression can be rescued with a proteasome inhibitor in two rhabdoid tumor cell lines. Finally, we also changed and improved our fractionation protocol to separate the nucleus and the nuclear soluble from the chromatin soluble to further verify that this was not simply a distributive mechanism. We thank this reviewer for this comment because it provides additional evidence for the overall drug mechanism.

5. 3E need to show action of MITHRA on U2OS control cells.

In this revision, we now include the effect of mithramycin on H3K27me3 in U2OS control cells. Indeed, we show that the increase in H3K27me3 is not seen in U2OS cells. However, if we silence

SMARCB1 using siRNA in U2OS to mimic SMARCB1-depleted rhabdoid tumor, we can induce H3K27me3 accumulation with mithramycin consistent with this effect being related to SMARCB1 deletion.

6. In figure 4, the authors need to show immunoblot confirming EZH2 knockdown. Also, EZH2 inhibitors can be used to confirm the protective effect against mithramycin.

Thank you for this comment, we have now included the data showing depletion of EZH2 with siRNA (Figure EV2D). Additionally, we also show the same effect with the EZH2 inhibitor tazemetostat (Figure 4E).

7. RNA seq - did total RNA change per cell indicating global changes in RNA production- were spike in controls used for the RNA-Seq.

The reviewer makes an excellent point and to be addressed in future studies with total RNA and spike-in controls. Spike in controls were not used for the RNA sequencing experiment. However, there is not a global effect evident in the data and for sure there is not a global reduction in expression as most of the genes are induced at 18 hours. To explore this more quantitatively, we analyzed the RNA-seq data with *quantro* to test for global differences between experimental samples (Hicks & Irizarry, 2015). The difference is significant ($p < 0.001$) but may be driven by the small variance within groups. Plotting the data (see Appendix S1) suggests a subtle, but consistent trend indicating lower production of RNA in post-treatment samples. Further, the high degree of variance across the 18-hour samples would suggest that cellular responses to mithramycin may differ, and as such, single-cell studies may identify determinants of response. However, single-cell studies were outside the scope of the current study and will be explored in the future.

8. Figure 5 legend DEF is labeled as 8 hours- I think you mean 18 hours.

Thank you for this comment. We have made the correction.

Referee #2 (Remarks for Author):

This is an interesting and provocative manuscript addressing an important biological and medical problem. However, the manuscript would benefit from the following improvements:

1) Experimental results of the screen in Fig 1A and B are unclear. Please show IC50 or AUC values.

Thank you for this comment. Cell line screens are frequently presented as relative sensitivity with, for example, the NCI-60 or PPTC. This is favored because when handling so many cell lines at once there is inherent noise in these assays that makes reporting of an absolute IC50 less favored. This is a fairly standard way to report this data. In this manuscript, we chose to report the data this way because of this noise and also because we were looking simply to describe the relative sensitivity of rhabdoid tumor compared to other cell lines. Therefore, we believe that the data representation which we have chosen is the best way to represent the data. Nevertheless, we agree with this reviewer that the current representation does not allow the reader to interpret the absolute effect of drug treatment in a more traditional way relative to other compounds. Therefore, in this revision, we selected 15 cell lines, including 7 rhabdoid cell lines, and determined the IC50 in a rigorous fashion with multiple biological, technical and experimental replicates per cell line. We have reported these IC50s in this revision and it is notable that the relative sensitivity is identical in these screens (See Appendix Table S1). Thank you for this comment, this addition certainly makes the manuscript more interpretable.

2) Phenotypic effects of EC8042 are intriguing. A more detailed evaluation is warranted: what are the effects of sub-IC50 concentrations on cell cycle kinetics and differentiation as measured by flow cytometry?

Thank you for this comment. We have now included the effect of the drug on cell cycle kinetics using flow cytometry as this reviewer suggested (See Figure EV1B-D). Interestingly, the compound shows a subtle dose dependent G2 arrest.

3) Is the apparent reduction in SMARCC1 and SMARCE1 chromatin association dependent on SMARCB1? In other words, what happens to the association of BAF subunits with chromatin as a function of EC8042 treatment upon re-expression of SMARCB1 in rhabdoid tumor cells. This will test the prediction that EC8042 regulates BAF complex assembly in SMARCB1-dependent manner in rhabdoid tumor cells, as opposed to secondary consequences of the drug effects that otherwise are independent of the dysregulated functions of the mutant BAF complexes in tumor cells.

4) Similarly, the apparent changes in H3K27 methylation upon EC8042 treatment can be related to the BAF complex by measuring this effect upon re-expression of SMARCB1 that has been found to affect Polycomb antagonism in these tumors.

Thank you very much for these comments. These have substantially strengthened this manuscript and linked the effects more definitively to SMARCB1-deficient SWI/SNF. These comments were also echoed in the comments of the other reviewers. In this revision, we show that the displacement of SWI/SNF from chromatin and accumulation of H3K27me3 by mithramycin indeed is dependent on SMARCB1 deficiency. We show that complementation with an inducible SMARCB1 (courtesy of the Roberts lab) no longer allows for displacement of SWI/SNF and does not lead to accumulation of H3K27me3 with mithramycin treatment when SMARCB1 is expressed (see Figures 3H, 3I, 4A, 4B). Further, we show that depletion of SMARCB1 in U2OS SWI/SNF wild-type cells also leads to the accumulation of H3K27me3, an effect not evident in the presence of SMARCB1 in this cell type (see Figure 4G). These experiments more definitively link the drug hypersensitivity to the defining molecular mutation of rhabdoid tumor.

5) What is known about the BAF regulation of SP1 expression in rhabdoid or other cell types? The manuscript should include a discussion whether this has been shown or not. Could SP1 gene expression changes be BAF-independent?

It has been previously shown that SWI/SNF interacts directly with SP1 at specific loci (Kadam & Emerson, 2003, Kadam, McAlpine et al., 2000). We have included this point in the manuscript. Thank you for the comment. However, the regulation of SP1 expression by SWI/SNF is a novel observation in the current manuscript. It is certainly possible that SP1 has an autocrine signaling loop as has been shown with many transcription factors. In addition, as this reviewer contends there are likely multiple mechanisms responsible for SP1 expression in a context dependent fashion. However, the exploration of these mechanisms is likely beyond the scope of this manuscript. Nevertheless, the relationship between SWI/SNF and SP1 expression is a novel and important observation for the field interested in mithramycin pharmacology.

6) RNA interference experiments need to include multiple independent siRNA constructs against EZH2, or alternatively genetic complementation rescue with EZH2 cDNA, given the recognized risk of off-target effects of RNA interference.

Thank you for the comment. It is notable that we used pooled siRNA consisting of multiple independent siRNA targeting EZH2 in an effort to minimize off target effects. In addition, in this

revision, we confirmed the effect using the EZH2 small molecule inhibitor tazemetostat which behaved in the same manner as the siRNA (see Figure 4E). Together, the data makes it unlikely that the effects are due to off-target effects of the siRNA.

7) Given the proposed model, it would be interesting to examine the effects of EC8042 in the presence of tazemetostat, which has been shown to regulate the BAF-PRC axis in rhabdoid tumors. If direct experiments cannot be done, then a discussion should be included, given that tazemetostat has recently been approved for patients.

Thank you for this comment. We have included the effects of tazemetostat in this revision. These experiments indeed demonstrate the contribution of EZH2 and the resulting accumulation of H3K27me3 to the cellular sensitivity of this tumor type to the drug (See Figure 4E).

8) The manuscript favors a model in which EC8042 regulates the BAF-PRC axis: do gene expression measurements support this model, e.g. does EC8042 exhibit significant changes in PRC gene sets? Are observed gene expression changes consistent with the morphologic mesenchymal differentiation that is observed?

Thank you for this comment. We did indeed find evidence of mesenchymal differentiation in our gene signature studies. We tried to highlight this in the first version. So in this revision, we did make this clearer. Please see figure 5. We identify both adipogenesis (NES 2.5; $P = 0.002$) and myogenesis (NES 1.82; $P = 0.002$). Interestingly, unpublished data from another investigator also is finding evidence of mesenchymal possible origins of this tumor. So it is likely that this finding will be validated in the future. The PRC2 gene set is a bit more challenging. I am not sure one would expect to find these data in our data because this tumor has altered PRC2 activity dependent on the mutation that is specific for this tumor. In addition, PRC2 activity is highly context dependent particularly for bivalent genes. Further, the majority of the studies in this tumor have focused on inhibiting PRC2. So we did not find PRC2 sets in an unbiased manner but I am not sure you would predict that you would. Instead, we show the dynamic between PRC2 and SWI/SNF at two specific loci in the genome and correlate binding with increased H3K27me3 and loss of gene expression at these loci.

9) Likewise, do EC8042-induced changes in chromatin accessibility occur at loci regulated by PRC or BAF?

Thank you for this comment, we did show a loss of binding of SWI/SNF and a gain of H3K27me3 at the identical loci in the genome using chromatin precipitation (see Figure 3). We showed a similar effect at the SP1 site (see Figure 5). Finally, we show changes in chromatin accessibility at sites described by others to be regulated by PRC2 and SWI/SNF in rhabdoid tumor (see Figure 6).

10) The xenograft effects are impressive. What plasma drug concentrations are achieved in this model, and how do these concentrations compare to the concentrations that induced chromatin and differentiation changes in vitro? Can this analysis inform the types of pharmacokinetic properties that an effective drug should have in human clinical trials?

Thank you for appreciating the xenograft data. This is the best xenograft data we ever generated in our lab. I would argue that this result by itself makes EC8042 a great clinical candidate for this tumor type with such striking and marked tumor regressions. We did not collect serum to measure drug levels since EC8042 has never been in humans and so there is no reference value. Having said that, all models would predict these concentrations should be achievable in patients in the clinic. In our mithramycin clinical trial, the drug accumulated to 17 nM in humans. In the

current study, the effects were observed at 75 nM EC8042. We have previously reported that the drug accumulates to 4300 nM in mice at <20% of the maximum tolerated dose. Further, allometric scaling in several species would predict that the drug should achieve serum levels over 10X higher than mithramycin or in excess of 170 nM. So this should be clinically relevant. But there is no way to know without doing the clinical study.

11) The manuscript would benefit from a discussion of specific mechanisms of action of EC8042, mithramycin and related aureolic acids, with respect to their molecular target(s), their selectivity for these targets, the potential for off-targets, and their relationship with the effects on tumor versus normal cells, particularly in developing tissues. This is a complex and confusing subject, and the new results reported in the manuscript, combined with the provocative model that is proposed, would benefit from this discussion.

Thank you for this comment and comment 12. Again, this comment is echoed in the other reviews. Therefore, we re-wrote the discussion and changed the nomenclature from gain of function to SMARCB1-deficient. In this revision, we highlight that the likely mechanism stems from the reduced affinity of SMARCB1-deficient SWI/SNF for chromatin. We include this in the discussion. It is possible that higher concentrations of mithramycin would displace non-SMARCB1 deficient SWI/SNF from chromatin thus impacting numerous cell types. Indeed, this may in fact explain the therapeutic window, although a dedicated study would be needed to establish this fact. However, the study does highlight that DNA binding compounds, long considered, non-specific can in fact be used in the proper cell context to inhibit critical targets such as ncSWI/SNF if a therapeutic window can be established.

12) This statement "The deletion or inactivation of SMARCB1 does not affect SWI/SNF structural integrity." is not supported by the published results. Please see papers by Kadoch et al and Roberts et al that show that the BAF complexes are structurally disrupted by loss of SMARCB1. Likewise, gain-of-function activities have been reported for mutant BAF complexes in synovial sarcoma cells. Residual complexes in rhabdoid tumor cells exhibit mostly loss-of-function phenotypes, e.g. loss of chromatin association. Please clarify the evidence that the SMARCB1-deficient residual BAF complexes associate with and regulate the expression of new genes or loci?

We agree that the gain-in-function nomenclature that we used in the previous version was not correct and inconsistent with conventions in the field. Our point about not affecting structural integrity is that the complexes still associate and are functional as has been established by glycerol gradient in multiple studies. Reviewer 3 had a similar concern with the gain of function nomenclature. Therefore, we changed the nomenclature to impact on SMARCB1-deficient SWI/SNF. Indeed with SMARCB1 deletion there is an alteration in SWI/SNF distribution and enrichment of co-localization of BRD9 and SMARCA4 referred to as a loss of function as this reviewer suggests. In our revision, we emphasize that the drug favors this altered distribution and its association with promoters and CTCF sites. Additionally, we show reversal of gene sets described by others that are associated with SMARCB1-deficiency.

13) Fig 5 axis is unclear: log fold change in which direction: drug/control or control/drug, please clarify

Thank you. We have made this correction, it is Drug/Control such that treatment causes an induction of gene expression.

Referee #3 (Remarks for Author):

This manuscript examines whether mithramycin, a putative SP1 transcription factor inhibitor, will prove effective as an inhibitor of growth of rhabdoid tumor cell lines. The authors have previously shown that mithramycin inhibits the growth of Ewing's sarcoma cell lines. Both types of tumors share defects in the SWI/SNF chromatin remodeling complex. Ewing's sarcoma expresses a fusion protein that knocks the SMARCB1 subunit out of the complex resulting in an abnormal oncogenic complex while rhabdoid tumors have lost SMARCB1 expression leading to a loss of 2 of the 3 normal complexes. In this report, the authors demonstrate that 3 out of 4 rhabdoid cell lines show sensitivity to therapeutically-relevant doses of mithramycin both in culture and in xenografts. They also demonstrate that the mechanism of inhibition does not involve the induction of DNA damage. Thus, mithramycin may provide an effective treatment for patients with rhabdoid tumors, a cancer with a generally dismal prognosis, with the potential for rapid translation into clinical trials. However, most of the other experiments in the manuscript raise serious concerns about the authors' interpretation of the results. Therefore, the mechanisms by which mithramycin inhibits the growth of rhabdoid cell lines remains unclear. The authors should address the concerns below in order establish how mithramycin inhibits the growth of rhabdoid tumors.

Major Comments:

1) Figure 1B- Apparently not all rhabdoid cell lines respond to mithramycin because the A204 rhabdoid cell line did not respond. This cell line was initially listed as an eRMS but was shown to be a rhabdoid tumor cell line in 2002 (PMID: 23882450). The authors should comment on this differential sensitivity. In addition, at least 10 MRT cell lines exist. Why do the authors only present mithramycin sensitivity data for 3 of them?

Thank you for the comment. In the original submission, we thought that 3 cell lines was probably representative of the disease. However, we agree that if we are going to claim a hypersensitivity, we probably should have included more lines. Therefore, in this revision, we obtained an additional 4 rhabdoid tumor cell lines bringing the total to 7. We evaluated the effect on viability in a rigorous fashion using multiple biological, technical and experimental replicates. We found that all of these cell lines showed at least equivalent sensitivity as the other rhabdoid tumor cell lines with G401 and A204 being the least sensitive to the drug. Interestingly, A204 is a bit of an outlier as every other rhabdoid tumor cell line is more sensitive to the drug.

2) The X axis labelling in Figure 1C is confusing. If this is log concentrations, why does the axis say nM?

The X-axis is the log of the concentration in nM. Perhaps the drug name was making it confusing. So we changed it to "log concentration (nM)"

3) The authors should provide references for MYT1 and CCND1 as "well-established SWI/SNF binding sites". Only a limited number of groups have found CCND1 as a direct target of the SWI/SNF complex. RNA-seq data has found few, if any, consensus targets for rhabdoid tumors, let alone SWI/SNF-mutant tumors.

This is a fair comment. These are binding sites reported by others that we had previously confirmed by ChIP in our lab. We changed the description to "binding sites established by others."

4) If mithramycin competes with the SWI/SNF complex for binding to the minor groove of DNA, why wouldn't it evict SMARCC1 and SMARCE1 from chromatin in the U2OS cell line in Figure 3? Are the authors proposing that only the ncBAF complex binds to the minor groove of DNA? If so, then mithramycin should still evict BRD9 and GLTSCR1/GLTSCR1L1 from chromatin in the U2OS cell line, an easy experiment to perform. Otherwise, the results in Figure 3 do not make sense.

Thank you for this comment. We have added BRD9 to all of the chromatin fractionations and indeed this complex does show less binding to chromatin in all cell lines tested following mithramycin treatment.

5) The authors provide no protein validation for any of their gene expression data, even for the knockdown experiments. This raises serious concerns about the validity of their conclusions.

Thank you. This was an oversight. We always confirm silencing by both qPCR and western blot for both siRNA and drug studies prior to sequencing. We have included the protein data in this revision (see Figure EV2D and EV2E).

6) Figure 4- Multiple groups have shown that MRT cell lines are sensitive to EZH2 knockdown or inhibitors due to the restoration of p16INK4A expression. Surprisingly, these authors observe no effect on growth of their MRT cell lines after knockdown of EZH2. How do they explain this discrepancy with previous studies? It was also unclear why cells treated with mithramycin for 16 hours would become irreversibly committed to growth arrest. Removal of the drug should restore the ability of the SWI/SNF complex to bind to chromatin and evict polycomb complexes. Therefore, it seems like mithramycin effects should be easily reversed after 16 hour exposure.

We do indeed observe an effect on proliferation with EZH2 knockdown and inhibitors. There are two reasons this was not obvious in the first version (1) the cells arrest, enlarge and flatten. We reported percentage confluence using our Incucyte microscope that quantitates the amount of the plate surface that is covered with cells and so the enlargement off-sets the arrest to some degree. We originally planned to add in data using fluorescently labeled nuclei that indeed showed the failure to proliferate with EZH2 silencing or blockade. However, this seemed irrelevant to this manuscript as it has been shown by multiple groups so we removed the data. (2) interestingly, the second reason is that the effects of EZH2 blockade take longer relative the effects observed here. This has also been observed by others and with other epigenetic targeting approaches. But again, these results are irrelevant to the current study and our data do not contradict these findings. Therefore, we simply removed the data in this revision.

In regards to the second point, we believe that the effect becomes irreversible because we trigger an epigenetic switch similar to an irreversible lineage committed differentiation switch. Presumably this is because the lower affinity SWI/SNF can't displace PRC2 at the sites that trigger the effect. Nevertheless, the data shows it is not reversible beyond this threshold exposure.

7) If mithramycin suppresses expression of SP1, why doesn't it show up in either the volcano plots in Figure 5 or in the list of genes in Table S1?

SP1 does not show up on the volcanoes because the Log FC cutoff was 1.5 and 2 and SP1 showed a Log FC of 1.1.

8) The ATAC-seq and H3K27ac ChIP-seq data in Figure 5 and in the genome tracks throughout the manuscript are different than other published reports. Why do the authors observe an uneven ATAC-seq signals around TSSs? Normally the signal is symmetrical around the sites (see the Pan et al. PMC6755913 for examples). In addition, H3K27ac usually appears as a dual peak at enhancers and TSSs. Some of the tracks show this common pattern but the heat maps and most of the tracks show a single wide peak. The fact that the authors do not list the program they used to plot the ChIP/ATAC-seq heatmaps makes it difficult to understand these differences.

We'd like to thank the reviewer for the thorough investigation of the ATAC and ChIP-seq tracks as well as point us to relevant examples in the literature with which to compare our results. As such, we have added the excellent Pan et al. manuscript to the list of citations for our results. Further, we have added additional methodological information describing the software utilized to generate the heatmaps (deeptools v3.4.3). The right-skewed heatmaps were the result of a misspecification of the reference point during the plotting of the TSS signal density and has since been amended and corrected in this revision (see Figure 6A and Appendix S3), which now reflects the expected symmetric distribution of ATAC-seq signal about active TSS sites. We agree with the reviewer that active promoters and enhancers will often display H3K27Ac marks with a characteristic peak-valley-peak motif, of which we observe this motif across sites in our ChIP-seq data. To further probe aspects of our results in the context of primary tissue chromatin states, we employed an 18-state chromHMM model derived from the TARGET cohort of primary malignant Rhabdoid tumor samples (Figure 6C), which offers superior specificity for this disease model and possibly even clinical relevance for the results presented, as opposed to relying solely on interpretation from peaks/peak shape derived from a single chromatin mark. The chromHMM model combined with our H3K27Ac ChIP-seq/ATAC-seq/RNA-seq data, we believe provides a richer contextual view of the resulting chromatin changes associated with MMA treatment when coupled with primary disease chromatin state information.

9) The authors refer to the ncBAF complex that remains in the rhabdoid tumors showing a "gain-of-function activity". In the Discussion, they refer to an "oncogenic SWI/SNF" in rhabdoid tumors as opposed to the "wild-type complex in U2OS cells". This is an incorrect interpretation of the papers from the Kadoch laboratory. The ncBAF complex normally appears in all cells. Because it does not possess SMARCB1 as a subunit, SMARCB1-deficient rhabdoid tumors lose the BAF and PBAF complexes but retain the ncBAF complex. Thus, the absence of the other 2 SWI/SNF complexes allows ncBAF to localize to sites normally occupied by BAF and PBAF, not a gain of function or mutation in the ncBAF members.

We very much appreciate this comment. "Gain of function" was certainly the wrong choice of words and not consistent with nomenclature in the field that refers to the alteration in SWI/SNF in synovial sarcoma as a "gain of function" and in rhabdoid tumor as a "loss of function". But as this reviewer states, "loss of function" is not altogether accurate either because there is a clear altered distribution of SWI/SNF and co-enrichment of BRD9 and SMARCA4 that favors promoters and CTCF sites. We wanted to emphasize that we were targeting the "oncogenic" activity of SWI/SNF which led to the poor word choice. Since we do not interrogate the three complexes in this study, we have changed the nomenclature to SMARCB1-deficient SWI/SNF as that includes all three complexes in this cell type. We clearly show that the resulting chromatin remodeling favors these promoter sites and CTCF. But, thank you for this comment.

Minor Comments:

The authors should stain the cells in Figure 2 E & G with Oil Red to demonstrate lipid deposits.

Thank you. We have included these data in the revision (see Figure 2H).

In summary, we believe we have addressed all of the comments of the reviewers. We look forward to hearing the response to this revision.

4th Nov 2020

Dear Dr. Grohar,

Thank you for the submission of your revised manuscript to EMBO Molecular Medicine. Thank you also for working with the journal and for re-supplying Figures 2, 3, 4 and 5. We have checked these images and are satisfied that there are no image aberrations. Furthermore, we have received the enclosed reports from the 2 referees who re-reviewed your manuscript. While referee #3 still has some concerns that should be addressed, they are both supportive of publication, and I am thus pleased to inform you that we will be able to accept your manuscript pending the following final amendments:

1) Referee #3's comments:

Please include the SMARCC1 ChIP-seq data mentioned in point 1. Please address the other referee's concerns in writing.

2) Main manuscript text:

- Please answer/correct the changes suggested by our data editors in the main manuscript file (in track changes mode). This file will be sent to you in the next couple of days. Please use this file for any further modification.
- Please provide up to 5 keywords.
- Please remove the colored and the strikethrough text.
- Remove "data not shown". As per our guidelines on "Unpublished Data" the journal does not permit citation of "Data not shown". All data referred to in the paper should be displayed in the main or Expanded View figures. "Unpublished observations" may be referred to in exceptional cases, where these are data peripheral to the major message of the paper and are intended to form part of a future or separate study, the names of the persons that reported the observation should be listed in brackets.
- The Material and Methods section should follow the Discussion, before the Acknowledgements. "Works cited" should be renamed "References".
- Please carefully check whether all figures/figure panels are referenced in the main manuscript text (Fig. 4D, Appendix Fig. S1 are not called out in the text).
- Material and methods: We note that that you refer to previously published methods. To ensure reproducibility, please make sure that enough information is available in your manuscript to reproduce the experiments. Please provide antibody dilutions and PCR primers. Include a mouse section indicating the origin, gender and age of the mice, as well as their housing and husbandry conditions. The statistical paragraph should reflect all information filled in the Authors checklist.
- Please indicate in the figures or in the legends the exact $n=$ and exact $p=$ values, not a range, along with the statistical test used. Some people found that to keep the figures clear, providing a supplemental table with all exact p -values was preferable. You are welcome to do this if you want to.
- Thank you for providing a data availability section. Please note that the data have to be made publicly available before acceptance of the manuscript.

3) Figures: Please make sure that scale bars are included in your pictures and defined in the legends.

Thank you for providing source data. Please upload them so as to have one pdf file per figure.

4) Appendix: Please merge the different appendix files to have one final file, including a Table of Content. The corresponding legends should be included in the appendix file, and removed from the main manuscript file. All appendix figures should be labelled and referenced as Appendix Fig. Sx.

5) In the checklist, please:

- Indicate the manuscript number on the top left corner
- Section B/2: if no inclusion/exclusion criteria were established, please add a sentence to that effect.
- Section D/8: indicate the origin of the mice, as well as their housing and husbandry conditions.

6) The paper explained: EMBO Molecular Medicine articles are accompanied by a summary of the articles to emphasize the major findings in the paper and their medical implications for the non-specialist reader. Please provide a draft summary of your article highlighting

7) For more information: There is space at the end of each article to list relevant web links for further consultation by our readers. Could you identify some relevant ones and provide such information as well? Some examples are patient associations, relevant databases, OMIM/proteins/genes links, author's websites, etc...

8) Every published paper now includes a 'Synopsis' to further enhance discoverability. Synopses are displayed on the journal webpage and are freely accessible to all readers. They include a short stand first (maximum of 300 characters, including space) as well as 2-5 one-sentences bullet points that summarizes the paper. Please write the bullet points to summarize the key NEW findings. They should be designed to be complementary to the abstract - i.e. not repeat the same text. We encourage inclusion of key acronyms and quantitative information (maximum of 30 words / bullet point). Please use the passive voice. Please attach these in a separate file or send them by email, we will incorporate them accordingly.

9) Please also suggest an image or visual abstract to illustrate your article as a png file 550 px-wide x 400-px high.

10) As part of the EMBO Publications transparent editorial process initiative (see our Editorial at <http://embomolmed.embopress.org/content/2/9/329>), EMBO Molecular Medicine will publish online a Review Process File (RPF) to accompany accepted manuscripts.

This file will be published in conjunction with your paper and will include the anonymous referee reports, your point-by-point response and all pertinent correspondence relating to the manuscript. Let us know whether you agree with the publication of the RPF.

I look forward to receiving your revised manuscript.

Yours sincerely,

Lise Roth

Lise Roth, PhD
Editor
EMBO Molecular Medicine

To submit your manuscript, please follow this link:

Link Not Available

Photos 400-800 DPI

*Additional important information regarding figures and illustrations can be found at <https://bit.ly/EMBOPressFigurePreparationGuideline>

The system will prompt you to fill in your funding and payment information. This will allow Wiley to send you a quote for the article processing charge (APC) in case of acceptance. This quote takes into account any reduction or fee waivers that you may be eligible for. Authors do not need to pay any fees before their manuscript is accepted and transferred to our publisher.

***** Reviewer's comments *****

Referee #2 (Remarks for Author):

I thank the authors for their thoughtful considerations and look forward to having this work published.

Referee #3 (Comments on Novelty/Model System for Author):

The authors have done an excellent job of addressing the multiple comments of the previous review. As in the previous submission, the most striking and impactful data come from the in vivo studies where the authors demonstrate a complete inhibition of rhabdoid tumor growth in a xenograft model. The fact that inhibition of growth occurred after only a 3 day continuous infusion of the drug further strengthens this novel finding. The authors also clearly exclude the current model for the action of mithramycin, inhibition of SP1 activity, as the cause of the cell line sensitivity. Nonetheless, despite the extensive molecular analyses provided in the manuscript, it still remains unclear why rhabdoid cell lines show a high sensitivity to mithramycin. While silencing of CCND1 could underlie the growth inhibition after mithramycin exposure, an experiment showing a reversal of this effect after expression of a CCND1 expression vector was not included. The authors also did not discuss the expression of CDKN2A (p16), the well-established inducer of growth arrest after SMARCB1 expression in rhabdoid cell lines. The growth inhibition could also arise from the induction of the cellular differentiation programs shown in the manuscript. In addition, the authors do not discuss why mithramycin would cause such a limited and specific eviction of the ncSWI/SNF complex from chromatin. However, the high significance of the xenograft studies along with the insights into mithramycin effects on chromatin structure and gene expression should prove of interest to the broad readership of this journal. The authors could improve the impact of this manuscript by addressing the comments below.

Major Comments:

1) According to the Material and Methods, the authors carried out ChIP-seq for SMARCC1, a key component of the SWI/SNF complex. The authors propose that selective eviction of the complex from chromatin represents a major effect of mithromycin treatment. However, they do not include a heatmap of SMARCC1 binding before and after mithramycin treatment, at least at the limited number of genes with altered expression, to confirm this mechanism. In addition, the inclusion of SMARCC1 binding tracks in all the figures showing binding of H3K27ac and chromatin accessibility at genes would also considerably strengthen the authors' conclusions.

Minor Comments:

- 1) The authors need to correct the multiple grammatical errors caused by the editing of the original manuscript.
- 2) The gene names in the volcano plots in Figure 4 are too small to read.
- 3) The authors could add more detail to the figures to make them reader-friendly. For example, the heatmaps in Figure 6A could be labelled as ATAC-seq to inform the reader about their identity.

Referee #3 (Remarks for Author):

The authors have definitely improved the quality of the manuscript. However, it was difficult to review given that none of the figures were labelled. Furthermore, the lack of information throughout the figures made it frustrating for interpretation of the data. I also am concerned that the authors do not fully comprehend the results of the multi-omics data, even though several of the investigators have clear expertise in this area. Nonetheless, the translation of these results to the clinic appears high. Along with the insights into some aspects of the underlying chromatin biology, the manuscript warrants publication in EMBO Molecular Medicine. However, they need to include

the SMARCC1 ChIP-seq data.

18th Nov 2020

Dear Patrick,

Thank you for submitting your revised manuscript to EMM and for addressing the editorial issues. Before final acceptance of your manuscript, we would need the following:

1/ Please include a point-by-point rebuttal letter regarding referee #3's comments (regarding SMARCC1 ChIP-seq).

2/ Please accept all changes in the main manuscript text.

3/ Thank you for providing a synopsis. I slightly modified it to match our style and format, please let me know if you agree with the following:

EC8042 is identified as an inhibitor of oncogenic SWI/SNF and a promising therapeutic candidate for rhabdoid tumor. The mechanism of target inhibition is elucidated and used to optimize compound administration, characterize an associated biomarker, and cure mice bearing rhabdoid tumor xenografts.

- Mithramycin displaced SMARCB1-deficient SWI/SNF from chromatin.
- Mithramycin induced epigenetic reprogramming favoring promoters and SWI/SNF bound CTCF sites to induce mesenchymal differentiation.
- EC8042, a second-generation mithramycin analogue, induced mesenchymal differentiation to cure 3 of 8 mice bearing rhabdoid tumor xenografts with a single 3-day infusion of the drug.

Thank you also for providing a nice picture. Please resize it as a 550x400 px picture, and make sure the text is still readable when resized.

4/ Please let us know whether you agree or not with the publication of the RPF.

Thank you for bearing with these last changes,

Looking forward to receiving your final manuscript,

With my best wishes,

Lise

Lise Roth, PhD
Editor
EMBO Molecular Medicine

Thank you for considering our article for publication in EMBO Molecular Medicine, “Mithramycin induces promoter reprogramming and differentiation of rhabdoid tumor”. Please find our second revision attached for **EMM-2020-12640-V3**. In this study, we establish a cellular sensitivity of rhabdoid tumor to mithramycin and its second-generation analog EC8042. We link this hypersensitivity to the defining lesion of the tumor, SMARCB1 deletion and establish the mechanism of action. We show promoter reprogramming, link the reprogramming to SMARCB1-deficient SWI/SNF and demonstrate reversal of the oncogenic phenotype and mesenchymal differentiation of the tumor both in vitro and in vivo. The net result is striking activity and complete cures of mice bearing rhabdoid tumor xenografts with a single 3-day infusion of EC8042. These data establish EC8042 as a clinical candidate for this tumor.

Thank you for the opportunity to resubmit these revisions. Please find our most recent responses below:

1/ Please include a point-by-point rebuttal letter regarding referee #3's comments (regarding SMARCC1 ChIP-seq).

Thank you for this comment. We agree that the previous version was not clear. We did not do CHIP sequencing of SMARCC1. Instead, we did CHIP-PCR of SMARCC1 at specific loci in the genome. We agree that the first version was not clear in the methods section and listed the CHIP qPCR of SMARCC1 in the section entitled Chromatin Immunoprecipitation with high throughput sequencing. In this revision, we have clarified that, and have separated out the CHIP qPCR of SMARCC1, H3K27me3 and H3K27ac from the ChIP sequencing of H3K27ac in the methods section of the manuscript. In addition we have specified all of the conditions for both sets of experiments. Thank you for noticing that this section was unclear.

2/ Please accept all changes in the main manuscript text.

Thank you we have accepted all of the changes it and labeled it “Clean”. We have now deleted the tracked changes submission from the website so the only version uploaded is the one with all of the changes accepted.

3/ Thank you for providing a synopsis. I slightly modified it to match our style and format, please let me know if you agree with the following:

EC8042 is identified as an inhibitor of oncogenic SWI/SNF and a promising therapeutic candidate for rhabdoid tumor. The mechanism of target inhibition is elucidated and used to optimize compound administration, characterize an associated biomarker, and cure mice bearing rhabdoid tumor xenografts.

- Mithramycin displaced SMARCB1-deficient SWI/SNF from chromatin.

- **Mithramycin induced epigenetic reprogramming favoring promoters and SWI/SNF bound CTCF sites to induce mesenchymal differentiation.**
- **EC8042, a second-generation mithramycin analogue, induced mesenchymal differentiation to cure 3 of 8 mice bearing rhabdoid tumor xenografts with a single 3-day infusion of the drug.**

Thank you. We agree with these edits.

Thank you also for providing a nice picture. Please resize it as a 550x400 px picture, and make sure the text is still readable when resized.

Please find the resized image in the new resubmission replacing the previous figure.

4/ Please let us know whether you agree or not with the publication of the RPF.

We agree with the publication of the RPF with the single sentence alluding to the imaging irregularities.

Thank you again for allowing us to re-submit this article. We look forward to a response.

20th Nov 2020

Dear Patrick,

I am pleased to inform you that your manuscript is now accepted for publication and will be sent to our publisher to be included in the next available issue of EMBO Molecular Medicine!

Congratulations on your interesting work,

With my best wishes,

Lise

Lise Roth, Ph.D
Scientific Editor
EMBO Molecular Medicine

Follow us on Twitter @EmboMolMed
Sign up for eTOCs at embopress.org/alertsfeeds

*** ** IMPORTANT INFORMATION ** **

SPEED OF PUBLICATION

The journal aims for rapid publication of papers, using using the advance online publication "Early View" to expedite the process: A properly copy-edited and formatted version will be published as "Early View" after the proofs have been corrected. Please help the Editors and publisher avoid delays by providing e-mail address(es), telephone and fax numbers at which author(s) can be contacted.

Should you be planning a Press Release on your article, please get in contact with embomolmed@wiley.com as early as possible, in order to coordinate publication and release dates.

LICENSE AND PAYMENT:

All articles published in EMBO Molecular Medicine are fully open access: immediately and freely available to read, download and share.

EMBO Molecular Medicine charges an article processing charge (APC) to cover the publication costs. You, as the corresponding author for this manuscript, should have already received a quote with the article processing fee separately. Please let us know in case this quote has not been received.

Once your article is at Wiley for editorial production you will receive an email from Wiley's Author Services system, which will ask you to log in and will present you with the publication license form for completion. Within the same system the publication fee can be paid by credit card, an invoice, pro forma invoice or purchase order can be requested.

Payment of the publication charge and the signed Open Access Agreement form must be received before the article can be published online.

PROOFS

You will receive the proofs by e-mail approximately 2 weeks after all relevant files have been sent to our Production Office. Please return them within 48 hours and if there should be any problems, please contact the production office at embopressproduction@wiley.com.

Please inform us if there is likely to be any difficulty in reaching you at the above address at that time. Failure to meet our deadlines may result in a delay of publication.

All further communications concerning your paper proofs should quote reference number EMM-2020-12640-V4 and be directed to the production office at embopressproduction@wiley.com.

Thank you,

Lise Roth, Ph.D
Scientific Editor
EMBO Molecular Medicine

Corresponding Author Name: Patrick Grohar, MD/PhD

Manuscript Number: EMM-2020-12640-V2